# Ricci-Filtration: Boosting Retrieval-Augmented Generation Reranking for Question-Answering Tasks with Discrete Ricci Flow

## Abstract

Ricci flow (Hamilton, 1982) is a curvature-guided diffusion process that deforms space by shrinking regions of high positive curvature and expanding those with negative curvature. Similarly, discrete Ricci flow on weighted graphs (Ni et al., 2019) modifies edge weights by shrinking edges with positive Ricci curvature and stretching those with negative Ricci curvature, effectively increasing the separation between clusters. Inspired by these two cornerstone works, we propose a geometry-based RAG reranker enhancement procedure called Ricci-Filtration. By modeling the input query and initial retrieved chunks as a network, where the input query and chunks serve as nodes and embedding-based pairwise relations define an initial graph, Ricci-Filtration leverages discrete curvature and Ricci flow to compute a graph-dependent filtration score for each query–chunk edge. The system first filters the initial chunks using this score; then, a reranker processes the remaining chunks to enhance generative performance. We provide a stylized theoretical analysis showing that normalized discrete Ricci flow can separate edge types on idealized community graphs, offering support for the post-flow filtering mechanism while not implying guarantees on arbitrary embedding-derived retrieval graphs. Experiments across QA benchmarks show that Ricci-Filtration improves several settings, especially SQuADv2 and selected MultiHop-RAG query types, while also revealing limitations on harder connected multi-hop reasoning tasks. These results establish the main contribution: a modular geometric prefilter that adaptively changes the context seen by an otherwise unchanged reranker and generator, yielding consistent SQuADv2 gains across two generators and substantial gains for selected MultiHop-RAG query types. Ablation studies characterize sensitivity to graph-construction thresholds, flow iterations, embeddings, rerankers, and a simple K-means filtering baseline.

Additional matched-budget and diagnostic pilots clarify how this contribution operates. They show that post-flow weights act as graph-structural context scores, with query information entering the released unit-initialized design primarily through graph construction. Direct relevance scores preserve annotated evidence more consistently in the small matched pilot, whereas Ricci-Filtration produces distinct adaptive context sets and the best matched-generation result on MuSiQue. These diagnostics complement rather than replace the main end-to-end results, supporting Ricci-Filtration as a useful geometric context-shaping stage and motivating similarity-aware initialization and faster flow approximations.

## 1 Introduction

Retrieval-augmented generation (RAG) (Lewis et al., 2020) is a popular method for using LLMs to answer queries based on data that is too large for the context window of a language model, which means the maximum number of tokens that can be processed by the LLM at once (Liu et al., 2024). A typical RAG system is designed to retrieve a few records that are specifically relevant to the user's query and collectively small enough to fit within the language model's context window (Baumel et al., 2018; Laskar et al., 2020; Yao et al., 2017). This process allows the LLM to generate a response informed by specific, external information

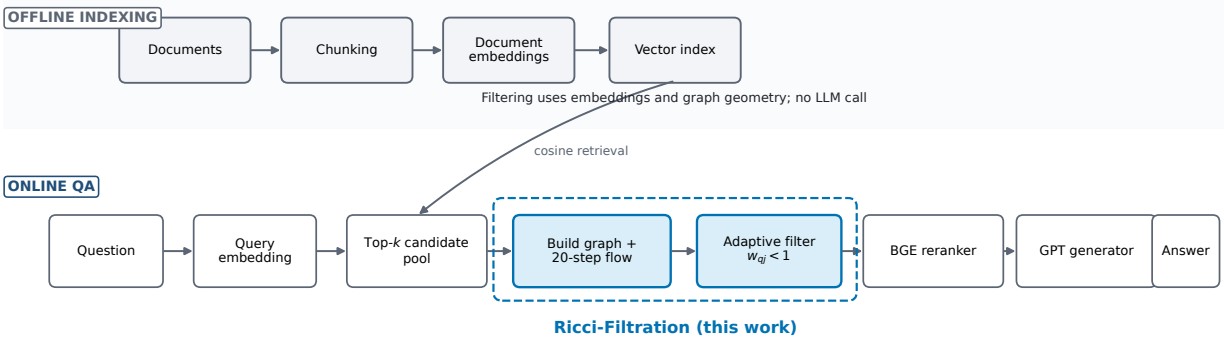

Figure 1: Overview of the offline indexing and online QA pipeline. The highlighted stages build the query–candidate graph, run the finite normalized flow, and apply the adaptive $w_{qj} < 1$ filter before a common BGE reranker and generator.

rather than just its pre-trained knowledge. However, the initial retrieval process may contain many irrelevant chunks, especially in Query-Answer (QA) tasks, which distracts the LLM and leads to hallucinations. More specifically, most RAG systems utilize vector search to select the most related chunks from the knowledge base, while vector search can be easily fooled by keyword overlap or shallow semantic similarity. One solution is to employ rerankers as fine-grained filters to not only move the most relevant "gold" chunks to the top, but also filter out "noisy" or irrelevant documents.

The paradigm described above was first proposed by Glass et al. (2022), where Cross-Encoders are employed as rerankers to perform a deep token-level comparison between the query and the initial selected chunks. Unlike bi-encoders where the query and document are mapped into a shared vector space independently, cross-encoders treat the query and document as a single combined input sequence by performing full self-attention across every token in both query and document simultaneously. Examples include transformer-based encoders like BGE-Reranker (Xiao et al., 2023) and Sentence-BERT (Reimers & Gurevych, 2019b). Since we have to input the query and the document into the model together, the latency can be high. Additionally, most cross-encoders are based on BERT-style architectures, which need to be fine-tuned and have token limits, making them less general in practice. There are also works that use LLMs as rerankers. Sun et al. (2023b) demonstrated that LLMs, when prompted to rank documents, significantly outperform smaller cross-encoders. However, the price would be the relatively high computational cost. On the other hand, the current reranker paradigm requires a fixed parameter $k$ to select reranked chunks. Such a one-size-fits-all setting may make the reranker less flexible to the variable relationships between query and text chunks. For instance, if $k$ is too large, the selected reranked chunks can still contain noisy information degrading the quality of the LLM's generation.

To address these issues, we introduce Ricci-Filtration, a RAG enhancement framework featuring a geometry-augmented reranking process based on the discrete Ricci flow (Ni et al., 2019). From the geometric point of view, the discrete Ricci flow defined on weighted graphs and deforms edges weights as time progresses. As a result, edges of large positive Ricci curvature will shrink and edges of negative Ricci curvature will be stretched, i.e., separated from each other. Unlike traditional rerankers, Ricci-Filtration relies exclusively on precomputed embeddings and the intrinsic geometric relationships between query and text chunks. By modeling the input query and initial retrieved chunks as a graph (network), where the input query and chunks serve as nodes and embedding-based pairwise relations define the initial graph topology, Ricci-Filtration evolves query-edge lengths and uses their post-flow values to make an adaptive graph-structural filtering decision; the downstream reranker then supplies the conventional semantic relevance stage. A reranker is then employed before generation. This approach is LLM-agnostic during the filtration step, since the filtering decision uses the initial embeddings and graph geometry rather than an additional LLM call. It therefore

adds an adaptive signal without an additional LLM filtration call, while the complete implementation retains an explicit accuracy–latency trade-off. Figure 1 illustrates the framework of Ricci-Filtration.

In the released unit-initialized design, the query embedding affects the graph topology and therefore the retained context. A fixed-topology diagnostic isolates this conditioning channel: once adjacency is artificially held constant, the forced query-edge magnitudes no longer carry query content. Appendix B quantifies this topology-conditioned behavior and evaluates similarity-aware initialization as a natural extension.

The main contributions of our work can be summarized as following:

1. We first model the retrieved candidate set from a graph-geometric perspective. Discrete Ricci flow then produces graph-structural query-edge scores and an adaptive retained count before conventional reranking. This adds a context-shaping signal that is complementary to pointwise relevance scoring.

2. To run the discrete Ricci flow on graphs, we construct an initial edge-incidence graph from cosine dissimilarities between the embeddings of the query and the initially retrieved text chunks. Present edges are then assigned positive initial lengths before the flow starts. This construction uses the retrieved embeddings and can be inserted before an existing reranker without changing the downstream generation model.

3. We give a stylized theoretical analysis showing edge-type separation under normalized Ricci flow. The uniform-neighborhood case $\alpha = 0, p = 0$ admits a closed-form proof, while the practical choice $\alpha = 1/2, p = 2$ admits a weaker finite-time separation guarantee on the same symmetric graph family. These results support the filtering mechanism but do not constitute a guarantee for arbitrary embedding-derived retrieval graphs.

4. Across two generators and multiple QA benchmarks, the complete Ricci-Filtration pipeline consistently improves all reported SQuADv2 metrics and improves overall and three of four MultiHop-RAG query types under Llama 3.1-8B-Instruct. Matched controls and diagnostics explain how this geometric signal differs from direct relevance ranking without overturning these end-to-end gains.

The remaining sections of this paper are organized as following schema. In section 2, we provide some preliminaries involving concepts of reranker and discrete Ricci flow. In section 3, we illustrate the architecture of Ricci-Filtration. The comparisons between RAG with Ricci-Filtration and RAG with other types of rerankers on benchmark datasets are provided in section 4. Ablation studies are provided in section 4 as well. In the end, section 5 discusses few characteristics and limitations of Ricci-Filtration and suggests some future directions. The code and instructions needed to reproduce the main experimental results are included in supplementary material.

## 2 Preliminary

### 2.1 Reranker in RAG

The RAG system consists mainly of two components: Retrieval and Generation. The retrieval process extracts relevant data from external sources via indexing and searching. Indexing organizes documents using inverted indexes for sparse retrieval or dense vector encoding for dense retrieval (Gao et al., 2023; Khattab & Zaharia, 2020; Douze et al., 2024; Yu et al., 2024a) to ensure efficiency. The searching phase then leverages these indexes to fetch documents based on user queries, often applying rerankers (Lyu et al., 2024; Tang & Yang, 2024) such as a powerful cross-encoder to refine the ranking of results for better relevance to the user query. The generation component utilizes the retrieved content and user query to formulate coherent and contextually relevant responses with the prompting and inference phases. LLMs are the preferred performance standard in generation and their dominance is attributed to their "Emerging" ability (Wei et al., 2022) and recent breakthroughs in following complex human commands (Ouyang et al., 2022). In this paper, we will focus on improving the reranking step which is a crucial second-stage in the retrieval process. In general, a reranker acts as a high-precision filter that double-checks the documents retrieved by our

initial search before sending them to the generation model. The common choices of rerankers involve cross-encoders and LLM-based rerankers. Cross-encoders process the query and the document simultaneously. Because they can perform "full attention" across both texts, they are highly accurate but computationally expensive (Reimers & Gurevych, 2019a; Devlin et al., 2019). On the other hand, LLM-based Rerankers use full LLM (like GPT-4o or Gemini) to rank retrieved documents. For example, we can prompt the LLM with the query and a list of docs, asking it to identify which are most relevant (Sun et al., 2023a; Yu et al., 2024b). However, the latency and cost will be extremely high.

Curvature-driven graph modification has also been studied in graph machine learning. In particular, Topping et al. (2022) introduce Stochastic Discrete Ricci Flow to rewire graph bottlenecks associated with over-squashing. That work changes graph topology to improve message passing in GNNs, whereas Ricci-Filtration keeps the retrieval graph topology fixed during the flow and uses the evolved query-edge lengths to select textual context. The shared use of curvature-guided surgery places the two methods in the same broad technique family, while their objectives and downstream tasks differ. RankRAG (Yu et al., 2024b) instruction-tunes one LLM jointly for context ranking and answer generation. However, Ricci-Filtration doesn't require the tuning of LLM.

## 2.2 Discrete Ricci Flow

Drawing on the geometric foundations established by Gauss and Riemann over 150 years ago, the Ricci flow utilizes curvature to provide a quantitative measure of how a space bends at any given point. In classical geometry, this curvature dictates the distribution of space: areas with high positive curvature are more "densely packed", whereas regions of negative curvature tend to spread out or expand. To locate these regions of large curvature, Hamilton (1982) introduced a curvature guided diffusion process, called the Ricci flow, that deforms the space in a way formally analogous to the diffusion of heat. Under the Ricci flow, regions in a space of large positive curvature shrink to points whereas regions of very negative curvature spread out. See Appendix E.4 for details. By considering a network (graph) as a discrete counterpart of a manifold and connected sum components as communities, Ni et al. (2019) introduced a discrete Ricci flow on networks for identifying communities in a network. Building on these foundations, we explore the application of discrete Ricci flow to the graphs established between user queries and retrieved document chunks in RAG.

To better illustrate the idea of discrete Ricci flow on weighted graphs, we follow the notations in Ni et al. (2019) and start with the definition of discrete (Ollivier) [1] Ricci curvature (Ollivier, 2007). Given a metric space $(X, d)$ equipped with a probability measure $m_x$ for each $x \in X$, for a given graph $G = (V, E)$ based on space $X$, let the edge weight of edge $xy \in E$ be $w_{xy}$, the discrete Ricci curvature of edge $xy$, $\kappa_{xy}$, is computed as follows:

$$\kappa_{xy} = 1 - \frac{\mathrm{W}(m_x^{\alpha,p}, m_y^{\alpha,p})}{d(x,y)}, \tag{1}$$

where W represents the optimal mass transport distance (a.k.a. Wasserstein Distance). The mass distribution $m_x^{\alpha,p}$ is defined as

$$m_x^{\alpha,p} = \begin{cases} \alpha & i = x \\ \frac{1-\alpha}{C} \cdot \exp(-d(x,i)^p) & i \sim x \\ 0 & \text{Otherwise}, \end{cases} \tag{2}$$

where $C = \sum_{j \sim x} \exp(-(d(x,j))^p)$, $d(x,i)$ is shortest path taken over all edge paths from $x$ to $i$, which is the induced metric from the edge weight. The notation $i \sim x$ represents node $i$ is the neighborhood of node $x$. More specifically, given a weighted graph $(V, E, w)$ with $w_{ij} > 0$ for all edges, the induced metric $d$ for edge $xy$ is defined by

$$d(x,y) = \min_{\gamma: x = v_0 \rightsquigarrow v_m = y} \sum_{r=0}^{m-1} w_{v_r v_{r+1}}, \tag{3}$$

---

[1] While the literature also includes Forman-Ricci curvature (Sreejith et al., 2016), which easier and faster to compute in large-scale networks, it lacks the geometric depth of Ollivier-Ricci curvature. Consequently, this work focuses exclusively on Ollivier-Ricci curvature, which we refer to simply as "Ricci curvature" throughout the remainder of the paper for brevity.

where the minimum is taken over all edge paths $\gamma$ from $x$ to $y$. It can be shown that the induced metric $d$ is well-defined.

In addition, the Wasserstein distance in (1) is defined as the minimum total weighted travel distance to move $m_x^{\alpha,p}$ to $m_y^{\alpha,p}$, i.e., $W(m_x^{\alpha,p}, m_y^{\alpha,p}) = \inf\{\sum_{u,v \in V} A(u,v)d(u,v)\}$ where $A$ is a discrete transport plan (map) from $V \times V$ to $[0,1]$ such that $A(u,v)$ is the amount of mass at vertex $v$ to be moved to vertex $u$ and $m_x^{\alpha,p}(u) = \sum_{v' \in V} A(u,v')$ and $m_y^{\alpha,p}(v) = \sum_{v' \in V} A(u',v)$. In the released experiments, this distance is evaluated through the GraphRicciCurvature implementation rather than by a direct CVXPY call. See Appendix E for more detailed explanations of theoretical concepts and optimal transport problem. Now the iteration process of discrete Ricci flow on network is defined as follows: The abstract metric definition above must be distinguished from the convention used by the released GraphRicciCurvature code. For an adjacent neighbor $i \sim x$, the implementation forms the kernel with the direct edge length $w_{xi}$, not the possibly shorter induced distance $d(x,i)$, and divides the transport cost by $w_{xy}$:

$$\widehat{m}_x^{\alpha,p}(i) \propto \exp(-w_{xi}^p), \qquad \widehat{\kappa}_{xy} = 1 - \frac{W_d(\widehat{m}_x^{\alpha,p}, \widehat{m}_y^{\alpha,p})}{w_{xy}}. \tag{4}$$

The optimal-transport cost matrix still uses all-pairs shortest-path distances induced by the current edge lengths. Direct edge length and induced distance can differ after an update, so the experimental algorithm uses $\widehat{\kappa}$ in (4); it should not be silently identified with the abstract metric curvature in (1).

$$w_{xy}^{(i+1)} = (1 - \varepsilon \kappa_{xy}^{(i)}) w_{xy}^{(i)}, \tag{5}$$

where $w_{xy}^{(i)}$ is the length of edge $xy$ at the $i$-th iteration and $0 < \varepsilon \le 1$ is the step size. Equation (5) is the abstract update; the released experimental implementation substitutes $\widehat{\kappa}_{xy}^{(i)}$ from (4), with shortest-path distances used inside the optimal-transport cost and direct edge length used in its denominator. See Algorithm 1 in section 3.

Note that the iteration (5) above will expand (stretch) edges with negative curvature and shrink edges with positive curvature. This process effectively causes intra-community edges (within-group connections) to condense and inter-community edges (between-group connections) to stretch. Because of this clear separation, a simple thresholding procedure can easily distinguish different communities. This is termed network "surgery" when edges of large weights (likely inter-community edges) are removed after several Ricci flow iterations.

## 3 Ricci-Filtration

By viewing the group of chunked texts and user query as a network, we propose Ricci-Filtration which further filters the initial selected chunks by their intrinsic geometric curvature with query. By definition (1), if two nodes $x$ and $y$ are from different communities (groups), their neighbor nodes tend to have fewer common neighbors, so the best way to move $m_x$ from $x's$ neighbors to $m_y$ in $y's$ neighbors has to travel along the edge $xy$ in most cases. As a result, the Wasserstein distance should be greater than the length of $xy$ ($d(x,y)$, leading to a negative Ricci curvature. Alternatively, nodes that are geometrically close or within the same community tend to share neighbors or have a shortcut between neighbors, thus the Wasserstein distance should not be greater than $d(x,y)$, leading to a positive Ricci curvature.

In the context of RAG, post-flow query-edge weights operationalize a graph-dependent partition of the candidate context before reranking. The method's end-to-end benefit does not require each weight to be a calibrated passage-relevance probability: the flow supplies a structural context signal, while the downstream reranker evaluates semantic relevance. We therefore use the precise terms "retained" and "filtered" for the two outcomes and report their evidence alignment separately in Appendix C. We may also get many byproducts such as the clusters of other text chunks after discrete Ricci flow. Since we only care about the relation between query and chunks, we ignore the byproducts like clustering between chunks in the current work. Exploring how to utilize the clusters formed purely from chunks is one of our future works. Algorithm 1 then shows how the discrete Ricci flow is implemented in practice.

The loop in Algorithm 1 conducts at most $M$ normalized discrete Ricci-flow iterations on a weighted graph. The post-flow step partitions the query-connected candidate chunks by removing high-weight query edges

---

**Algorithm 1** Finite-Time Discrete Ricci Flow

---

**Require:** An undirected graph $G$ formed by the query and candidate chunks. The maximum number of flow iterations $M$. The threshold value $\eta$ for post-flow filtering. The step size $0 < \varepsilon \leq 1$ and optional curvature-spread tolerance $\delta$. Set $w_{xy}^{(0)} = 1$ for every present edge and compute $\widehat{\kappa}_{xy}^{(0)}$ from (4).

1: **for** $i = 0, \ldots, M - 1$ **do**
2:     Update $\widetilde{w}_{xy}^{(i+1)} \leftarrow (1 - \varepsilon \widehat{\kappa}_{xy}^{(i)}) w_{xy}^{(i)}$.
3:     Normalize after the update: $w_{xy}^{(i+1)} \leftarrow \widetilde{w}_{xy}^{(i+1)} \cdot \frac{|E|}{\sum_{uv \in E} \widetilde{w}_{uv}^{(i+1)}}$.
4:     Compute the induced shortest-path cost matrix and the next implementation curvatures $\widehat{\kappa}_{xy}^{(i+1)}$ using (4).
5:     If $\max_{xy} \widehat{\kappa}_{xy}^{(i+1)} - \min_{xy} \widehat{\kappa}_{xy}^{(i+1)} < \delta$, stop early.
6: **end for**
7: For the query node $q$, keep chunk node $j$ only if the final normalized query-edge weight satisfies $w_{qj} < \eta$; otherwise remove it before reranking.
8: **Return:** The filtered chunk set and the finite-time weighted graph $G$.

---

after the iteration budget or optional early-stopping condition is reached. In the end, for edges connected with query and other chunks, we obtain the corresponding finite-time Ricci curvatures and edge weights, which can be used as heuristics for inter-community and intra-community edges. The normalization procedure rescales the edge weights after every update so that the average edge weight is 1; thus Algorithm 1 is a finite-time normalized Ricci flow. This update–normalize–recompute order and the strict cutoff $w_{qj} < 1$ match the released GraphRicciCurvature-based implementation (step size $\varepsilon = 1$, maximum $M = 20$, and default $\delta = 10^{-4}$). As theoretical context for the filtering mechanism, Theorem 3.1 gives a normalized adaptation of Ni et al. (2019) for community-structured graph families.

Related well-posedness results provide useful context but do not directly prove the exact practical algorithm used here. Bai et al. (2025) establish existence and uniqueness for a continuous-time normalized Ollivier Ricci-flow on weighted graphs, while Ma & Yang (2025) introduce modified and quasi-normalized Ricci flows on arbitrary weighted graphs with global existence and uniqueness. These results support the view that Ollivier-type Ricci-flow dynamics can be formulated rigorously on weighted graphs. However, Ricci-Filtration uses an embedding-derived edge-incidence graph, forced query edges, a finite number of discrete normalized iterations, and a heuristic post-flow surgery threshold $\eta = 1$. We therefore cite these works as theoretical context rather than as convergence, optimality, or correctness guarantees for Algorithm 1.

**Theorem 3.1.** *Take the complete* [2] *graph on $b+1$ vertices $p_1, ..., p_{b+1}$ and $b+1$ complete graphs $C_1, ..., C_{b+1}$ on $a+1$ vertices. Take a vertex $u_i$ from each $C_i$ and identify $u_i$ with $p_i$. The resulting graph is $G(a, b)$. Then the normalized Ricci flow associated to the Ollivier $K_0$-Ricci curvature detects the community structure on $G(a, b)$ if $a > b \geq 2$, namely, the weight of the intra-community edges shrink asymptotically faster than the weight of the inter-community edges, where the Ollivier Ricci curvature $K_0$ corresponds to $\alpha = 0, p = 0$ in equation* (1).

**Proposition 3.2** (Practical-parameter separation on $G(a, b)$)**.** *On the same graph family $G(a, b)$ with $a > b \geq 2$, let $d_1, d_2, d_3$ denote the edge lengths for gateway-to-gateway, gateway-to-non-gateway, and non-gateway-to-non-gateway edge types, respectively. If $d_1 \geq d_2 \geq d_3 > 0$ and the direct-edge node distribution $\widehat{m}$ in (4) uses $\alpha = 1/2, p = 2$, then the corresponding Wasserstein updates satisfy $W_1 > W_2 > W_3$ and*

$$\frac{W_3}{W_1} \leq q_{a,b} \frac{d_3}{d_1}, \qquad q_{a,b} := \frac{(a-1)(a+b)}{a(2a+b-1)} < 1.$$

*Consequently, starting from unit edge lengths and applying the unit-step normalized update, the non-gateway intra-community edge contracts geometrically relative to the inter-community edge.*

The proof of Theorem 3.1 utilizes the asymptotic behavior of a linear difference-equation system formed by different edge structures; see Appendix F. Proposition 3.2, proved in Appendix G, gives a weaker but

---

[2] A graph is complete if every distinct pair of vertices is connected by exactly one unique edge and it has no loops

more directly relevant analogue for the practical parameters used in our experiments. It provides idealized theoretical support for the separation mechanism observed empirically under the choice $\alpha = 1/2, p = 2$. It should be interpreted as a stylized explanation of why Ricci-flow filtering can separate edge types, rather than as a guarantee of empirical performance on embedding-derived retrieval graphs. Because the $(\alpha, p) = (1/2, 2)$ node measures depend exponentially on the current edge lengths, this practical-parameter extension is nonlinear and does not give the same closed-form eigenvalue proof as the uniform-neighborhood case in Theorem 3.1.

Together, Theorem 3.1 and Proposition 3.2 motivate the surgery rule by showing that, in idealized community graphs, normalized Ricci-flow updates can separate inter-community and intra-community edge types. Algorithm 1 uses a finite-time version of this idea: after at most $M$ normalized iterations, it applies a threshold to the query edges. In the normalized implementation, the value $\eta = 1$ should be read as a relative cutoff because Algorithm 1 rescales the edge weights so that their average is 1 at each iteration. Thus query edges with final normalized weights strictly below 1 are retained as within-community candidates, while query edges at or above 1 are filtered as stretched candidates. This threshold worked in our experiments, but it is a heuristic finite-time surgery threshold rather than a theorem-derived universal constant.

Ricci-Filtration deliberately uses a finite iteration budget: the filtering decision is made from the 20-step separation rather than requiring asymptotic convergence. In the 200-query mechanism pilot, every run reached the $M = 20$ cap and none met the optional curvature-spread tolerance $\delta = 10^{-4}$, so we refer to the outputs precisely as finite-time flow scores. This fixed budget makes the implemented decision rule reproducible and is consistent with the trajectory separation in Figure 3.

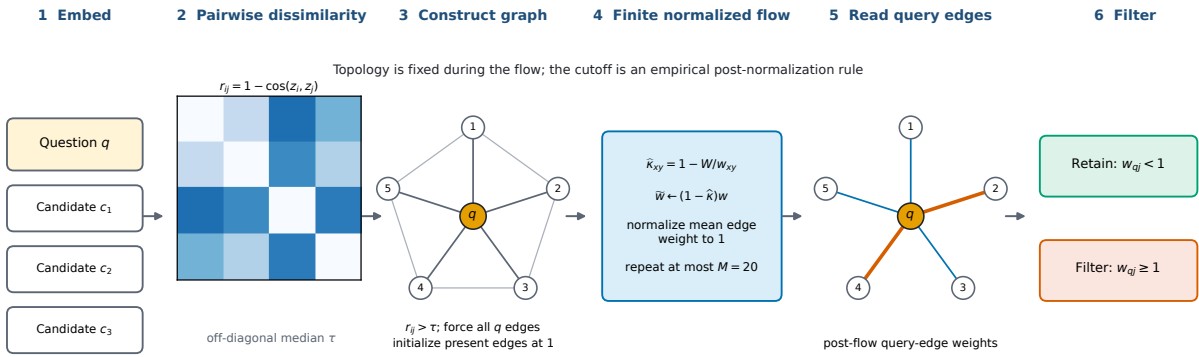

Figure 2: Schematic of the implemented filtration rule. Cosine dissimilarities define the graph topology, present edges start at unit length, implementation curvature is updated and normalized for at most $M = 20$ steps, and only final query edges with $w_{qj} < 1$ are retained. The two output groups are the adaptive context decisions passed to the common reranker.

## 3.1 Graph construction from embeddings

In Ricci-flow iteration, the neighborhood topology and the positive edge lengths play different roles. We therefore distinguish three objects: cosine dissimilarities between embeddings, a binary edge-incidence matrix defining the initial graph topology, and the positive edge lengths updated by Ricci flow. Let $z_i$ be the embedding of node $i$. For every pair of query or candidate chunk nodes, define the cosine dissimilarity

$$r_{ij} = 1 - \cos(z_i, z_j). \tag{6}$$

Let $\tau$ be a percentile threshold computed from the off-diagonal values of $r_{ij}$ among the query and initially retrieved candidate chunks. We define the binary edge-incidence matrix $B$ by

$$B_{ij} = \begin{cases} 1 & r_{ij} > \tau, \ i \neq j, \\ 0 & \text{otherwise.} \end{cases} \tag{7}$$

The entry $B_{ij} = 1$ means that the edge $\{i, j\}$ is present in the computational graph; it should not be interpreted as a claim of high semantic similarity. Since the threshold is applied to cosine dissimilarity, these edges include relatively dissimilar pairs that allow the Ricci flow to stretch cross-community relations. We additionally force $B_{qj} = B_{jq} = 1$ for every candidate chunk node $j$ so that every query-chunk edge is available after the flow. After these incidences are fixed, present query edges start with unit length rather than preserving the magnitude of the original query–chunk cosine similarity. The actual pipeline nevertheless remains query conditioned because the query embedding contributes to the global threshold $\tau$, determines graph adjacency, and participates in the transport neighborhoods. The query-swap diagnostic in Appendix B isolates these two channels by comparing frozen and recomputed topologies. The initial graph and initial edge lengths are

$$E_0 = \{\{i, j\} : B_{ij} = 1\}, \qquad w_{ij}^{(0)} = 1 \quad \text{for } \{i, j\} \in E_0. \tag{8}$$

Thus the value 1 in $B$ encodes edge incidence, whereas the value 1 in $w^{(0)}$ encodes the common initial edge length. This graph-construction threshold $\tau$ is different from the post-flow surgery threshold $\eta$. Because Algorithm 1 keeps the average edge weight at 1, we set $\eta = 1$ as a post-normalization cutoff for deciding which query-connected chunks remain after the flow. Edges with final normalized query-edge weights strictly below 1 are retained, while edges at or above 1 are filtered; this strict inequality matches the released implementation. More rigorous methods of defining both the initial graph topology and surgery threshold will be our future work.

The stylized theory supplies a mechanism for edge-type separation under normalized flow, and the practical-parameter proposition treats the same $\alpha = 1/2, p = 2$ choice used here on the idealized graph family. Query-swap and topology diagnostics complement that analysis by showing how the implemented retrieval graph receives query information and how performance changes with graph construction; full results appear in Appendix B.

### 3.2 Illustrative finite-flow trajectory

Figure 3 visualizes the 20-step query-edge trajectories for one frozen HotpotQA candidate graph using $\alpha = 0.5$, $p = 2$, and the median cosine-dissimilarity topology.

All present edges start at one. After the finite flow, query edges strictly below $\eta = 1$ are retained. The plot illustrates numerical separation under the implemented rule without asserting that a small final weight is a calibrated relevance probability.

## 4 Experiments

### 4.1 Datasets

To comprehensively evaluate the performance of Ricci-Filtration on general QA tasks, we follow the setting in Han et al. (2025) and select five widely used datasets that cover different perspectives. More specifically, we choose the Stanford Question Answering Dataset (SQuAD2.0) (Rajpurkar et al., 2018) and TriviaQA (Joshi et al., 2017) for the single-hop QA task. As to the multi-hop QA task, we select HotPotQA (Yang et al., 2018), MultiHopRAG (Tang & Yang, 2024), and MuSiQue (Trivedi et al., 2021) datasets. MuSiQue and TriviaQA are used in the further ablation studies. See detailed introduction of selected datasets in Appendix D. We use Accuracy, Precision (P), Recall (R), and F1-score as evaluation metrics for the SQuAD2.0 and HotPotQA datasets, and only accuracy is reported for the TriviaQA, MultiHop-RAG and MuSiQue.

Figure 3: Twenty-step query-edge trajectories for one illustrative candidate graph. The dashed line is the post-normalization decision cutoff; retained and filtered trajectories are filter outcomes, not verified semantic relevance labels.

Table 1: Evaluation results (%) on SQuAD2.0 and HotpotQA datasets with different reranker methods and generation LLMs. The mean metric value and standard deviation are based on 5 seeds.

| Methods | SQuAD2.0 | | | | | | | | HotpotQA | | | | | | | |
|---|---|---|---|---|---|---|---|---|---|---|---|---|---|---|---|---|
| | Llama 3.1-8B Instruct | | | | gpt-4o-mini | | | | Llama 3.1-8B Instruct | | | | gpt-4o-mini | | | |
| | P | R | F1 | Acc | P | R | F1 | Acc | P | R | F1 | Acc | P | R | F1 | Acc |
| No Reranker | 47.6±0.7 | 52.3±0.6 | 48.4±0.8 | 52.0±0.8 | 65.3±0.5 | 70.9±0.6 | 66.5±0.7 | 69.5±0.7 | 59.4±0.7 | 58.0±0.6 | 56.5±0.8 | 61.5±0.8 | 67.5±0.7 | 67.3±0.8 | 65.2±0.8 | 69.5±0.6 |
| Cross-Encoder Reranker | 47.7±0.8 | 52.1±0.7 | **48.5**±0.7 | 50.9±0.7 | 63.6±0.8 | 69.5±0.7 | 64.8±0.7 | 68.2±0.7 | **62.5**±0.8 | 59.6±0.8 | 58.9±0.9 | 64.0±0.7 | **69.7**±0.6 | **69.9**±0.6 | **67.5**±0.8 | **72.5**±0.8 |
| LLM-based Reranker | 37.9±0.7 | 41.2±0.6 | 38.3±0.6 | 49.5±0.8 | 37.5±0.8 | 42.6±0.8 | 38.6±0.7 | 52.1±0.7 | 62.5±0.7 | **60.6**±0.8 | **59.4**±0.7 | **64.4**±0.6 | 69.5±0.5 | 69.3±0.8 | 67.2±0.6 | 71.9±0.5 |
| **Ricci-Filtration** | **52.1**±0.7 | **57.3**±0.7 | **53.0**±0.8 | **55.4**±0.7 | **69.1**±0.8 | **73.6**±0.7 | **70.1**±0.7 | **73.0**±0.58 | 62.5±0.8 | 59.3±0.7 | 59.1±0.7 | 63.0±0.8 | 65.7±0.6 | 64.0±0.8 | 63.3±0.8 | 67.0±0.6 |

Table 2: Accuracy (%) on the MultiHop-RAG dataset across different query types with different reranker methods and generation LLMs. The mean accuracy and standard deviation are based on 5 seeds.

| Methods | LLama 3.1-8B Instruct | | | | | gpt-4o-mini | | | | |
|---|---|---|---|---|---|---|---|---|---|---|
| | Inference | Comparison | Null | Temporal | Overall | Inference | Comparison | Null | Temporal | Overall |
| No Reranker | 87.6±0.7 | 56.0±0.7 | 57.8±0.7 | 58.8±0.7 | 66.9±0.6 | 94.6±0.8 | 63.8±0.8 | 93.0±0.8 | 56.6±0.7 | 75.4±0.6 |
| Cross-Encoder Reranker | 90.9±0.8 | 56.7±0.7 | 60.8±0.6 | 57.5±0.8 | 68.3±0.7 | 96.7±0.7 | **66.0**±0.8 | 94.0±0.6 | **60.2**±0.6 | **77.8**±0.8 |
| LLM-based Reranker | 90.3±0.8 | 54.1±0.7 | 56.7±0.6 | 60.1±0.7 | 67.2±0.7 | 95.2±0.8 | 63.7±0.8 | 93.1±0.7 | 56.7±0.8 | **75.7**±0.6 |
| **Ricci-Filtration** | **93.9**±0.6 | **58.5**±0.8 | **83.4**±0.6 | **60.2**±0.7 | **73.1**±0.79 | **97.3**±0.6 | 64.1±0.8 | **95.1**±0.6 | 56.4±0.6 | 76.5±0.8 |

## 4.2 Evaluation settings

### 4.2.1 RAG

We employ cosine similarity based dense retrieval approach as our RAG method (Karpukhin et al., 2020), which is common in literature. Specifically, we first split the text into chunks, each containing approximately 256 tokens. For indexing, we use OpenAI's text-embedding-3-small model, which has demonstrated effectiveness across various tasks (Deng et al., 2025). For the standard RAG baseline (without a reranker), we retrieve the top five chunks based on cosine similarity. To generate responses, we utilize the open-source Llama-3.1-8B-Instruct model (Grattafiori et al., 2024). We also evaluate under gpt-4o-mini to test whether the observed filtering behavior depends on a single generator.

**Comparison protocol.** All methods start from the same top-20 dense-retrieval pool and use the same generator and scoring pipeline. Baselines send five chunks; Ricci-Filtration reranks and sends its dynamically

retained set, so the main tables compare complete pipelines rather than matched context budgets. Detailed construction, scoring, SQuADv2 sampling caveats, and the completed source-only protocol pilot are reported in Appendix A.

### 4.2.2 Reranker baseline

For comparison, we choose cross-encoder reranker and LLM-based reranker (Sun et al., 2023a) as the baseline methods. We select the top five reranked chunks from an initial pool of the top 20 similarity-based candidates. For time efficiency and cost, we employ open-source cross encoder named bge-reranker-base (Xiao et al., 2023), which is pre-trained on large-scale pairs data using contrastive learning. In ablation studies, we tried different cross encoders like MS-MARCO-MINILM-L6-V2 and MS-MARCO-MINILM-L12-V2, which are fine-tuned versions of the original MiniLM model(Wang et al., 2020) on MS MARCO task (Nguyen et al., 2017). As to the LLM-based reranker, we prompt gpt-4o-mini for reranking. We chose pointwise reranking because it provides clear scores that are easy to use and allows useful optimizations. The reranking prompt is given in Appendix J.

### 4.2.3 Ricci-Filtration

As suggested by Ni et al. (2019), we set $\alpha = 0.5, p = 2$ in the node distribution $m_x^{\alpha,p}$. For large $p$, the far neighbors of the source node will be heavily discounted. The released experiments compute Ollivier curvature and flow with the GraphRicciCurvature implementation, which uses exact optimal transport for the graph sizes considered here; the earlier statement that the experiment called CVXPY directly was inaccurate. We compute cosine dissimilarities from the embedded vectors and use them to construct the binary edge-incidence matrix in (7). A threshold is required to convert these pairwise dissimilarities into a graph topology. In the following experiments, we set $\tau$ to the 50th percentile of the off-diagonal cosine-dissimilarity values[3]. In addition, we force the query node to be connected with each chunk node in order to have all information of edges corresponding to query and chunks such as weights available after discrete Ricci flow iteration. Each present edge is initialized with unit length before the finite-time normalized flow. We use step size $\varepsilon = 1$, at most $M = 20$ iterations, the implementation's curvature-spread early-stopping tolerance $\delta = 10^{-4}$, and the strict post-flow cutoff $w_{qj} < 1$. To keep the comparison with the cross-encoder baseline controlled, we use the same bge-reranker-base cross-encoder after Ricci-Filtration; the difference is that the cross-encoder baseline reranks all 20 retrieved candidates, whereas Ricci-Filtration first removes chunks with high post-flow query-edge weights and then reranks the remaining subset.

Implementation details for the new mechanism pilots are provided in Appendix A.3.

### 4.3 Results

Table 1 compares different reranker methods on SQuAD2.0 and HotpotQA, where precision, recall, F1-score, and accuracy are reported. We also evaluate different generation LLMs to test whether the filtering behavior is tied to one generator. Following the standard practice in Han et al. (2025), we report accuracy for MultiHop-RAG in Table 2. The best performing metrics under different generation LLMs are highlighted in bold. The results show that Ricci-Filtration is most effective on SQuADv2 and on selected MultiHop-RAG query types. On SQuADv2, it improves every reported metric under both generation LLMs, with gains of 4.12–5.48 percentage points over the common cross-encoder baseline. On MultiHop-RAG with Llama 3.1-8B-Instruct, Ricci-Filtration improves overall accuracy from 68.27% to 73.12% and improves three of four query types; the largest gain is on null queries, from 60.80% to 83.39%. These multi-metric, cross-generator results support the paper's central selected-regime claim. The much smaller sub-one-point differences elsewhere should be treated as exploratory, but that uncertainty does not reduce the 4–23 point headline effects to marginal changes. The matched-budget pilot in Section C asks the separate mechanism-isolation question and should be read together with, not in place of, these complete-pipeline results. The overall results remain mixed across datasets and query types.

---

[3]The ablation tables report this graph-construction threshold as a percentile. For example, 50% means that $\tau$ is the median off-diagonal cosine dissimilarity, and a non-query pair is included when $r_{ij} > \tau$.

Notably, we observe that Ricci-Filtration has weakest performance on the HotpotQA benchmark across both generation LLMs, suggesting its limitation in multi-hop reasoning tasks where only critical and connected chunks should be used. However, current Ricci-Filtration tends to maintain excessive number of chunks after iteration, introducing noise that misleads the reranker during multi-step reasoning. Ablation studies on MuSiQue (Appendix H) also confirm this trend. There are clues that the margin of Ricci-Filtration might be diminished when strong generation LLMs are used, which is expected. However, these results suggest that geometric filtering may improve answer quality in some settings without using an LLM to make the filtering decision. This should be interpreted as a resource trade-off rather than a blanket deployment-cost claim, since the Ricci-flow iterations introduce measurable latency (Appendix K).

As to the LLM-based reranker, we see that its performance varies among different datasets. One reason is that the ability of LLM-based rerankers relies heavily on the underlying reranking LLM and reranking prompt. It is also sensitive to workflow design. Another drawback is the per-query API cost of scoring retrieved chunks. To keep the experiment bounded, we use gpt-4o-mini as the reranker model. We found that replacing it with stronger models can improve results, but at a substantially higher API cost. Compared with this LLM-based reranker, Ricci-Filtration avoids LLM calls during the filtering step, but its iterative graph computation still incurs latency, so the total deployment cost depends on hardware, implementation, and workload.

### 4.4 Matched-budget mechanism pilot

Approximately token-matched controls distinguish the geometric filter from ordinary relevance ranking. Cosine and BGE preserve annotated evidence more consistently, whereas Ricci's adaptive graph signal yields different context sets and the highest MuSiQue F1 in the matched generation pilot. This complementary behavior clarifies the role of Ricci flow as a structural context signal; it does not overturn the main end-to-end finding that the complete Ricci-Filtration pipeline benefits SQuADv2 and selected MultiHop-RAG settings. The complete design and results appear in Appendix C.

### 4.5 Ablation studies

Table 3 reports the extended TriviaQA parameter/model ablation; the MuSiQue ablation and K-means comparison remain in Appendices H and I.

Table 3: Ablation studies of Ricci-Filtration on TriviaQA accuracy (%).

| Methods | $k$ | $n^*$ | $\tau$ | $p$ | $M$ | Embedding model | Reranker model | Acc |
|---|---|---|---|---|---|---|---|---|
| Cross-Encoder | 20 | 20 | – | – | – | text-embedding-3-small | ms-marco-MiniLM-L12-v2 | 69.00 |
| Cross-Encoder | 20 | 10 | – | – | – | text-embedding-3-small | ms-marco-MiniLM-L12-v2 | 69.00 |
| Cross-Encoder | 20 | 5 | – | – | – | text-embedding-3-small | ms-marco-MiniLM-L6-v2 | 68.00 |
| Cross-Encoder | 20 | 5 | – | – | – | text-embedding-3-small | ms-marco-MiniLM-L12-v2 | 69.00 |
| Ricci-Filtration | 10 | 7.0 | 50% | 2 | 20 | text-embedding-3-small | ms-marco-MiniLM-L12-v2 | **72.00** |
| Ricci-Filtration | 20 | 16.7 | 50% | 2 | 20 | text-embedding-3-small | ms-marco-MiniLM-L12-v2 | **74.00** |
| Ricci-Filtration | 20 | 15.9 | 50% | 5 | 20 | text-embedding-3-small | ms-marco-MiniLM-L12-v2 | 70.00 |
| Ricci-Filtration | 20 | 13.5 | 50% | 1 | 20 | text-embedding-3-small | ms-marco-MiniLM-L12-v2 | 69.00 |
| Ricci-Filtration | 20 | 17.0 | 25% | 2 | 20 | text-embedding-3-small | ms-marco-MiniLM-L12-v2 | 69.00 |
| Ricci-Filtration | 20 | 16.0 | 75% | 2 | 20 | text-embedding-3-small | ms-marco-MiniLM-L12-v2 | 64.00 |
| Ricci-Filtration | 20 | 14.8 | 50% | 2 | 10 | text-embedding-3-small | ms-marco-MiniLM-L12-v2 | **71.00** |
| Ricci-Filtration | 20 | 10.4 | 50% | 2 | 1 | text-embedding-3-small | ms-marco-MiniLM-L12-v2 | 65.00 |
| Ricci-Filtration | 20 | 16.6 | 50% | 2 | 20 | text-embedding-ada-002 | ms-marco-MiniLM-L12-v2 | 66.00 |
| Ricci-Filtration | 20 | 16.7 | 50% | 2 | 20 | text-embedding-3-small | bge-reranker-base | **72.00** |
| Ricci-Filtration | 20 | 16.7 | 50% | 2 | 20 | text-embedding-3-small | ms-marco-MiniLM-L6-v2 | **74.00** |

*Note.* $n^*$ is the average number of chunks entering the reranker.

The ablation demonstrates that the proposed pipeline is effective across several configurations: the default $k = 20$, $\tau = 50\%$, $p = 2$, $M = 20$ setting reaches 74%, and changing the downstream reranker retains 72–74% accuracy. It also identifies meaningful design sensitivity: a moderate graph threshold and sufficient flow budget perform best, while the embedding model has a larger effect than the tested reranker substitutions.

## 5 Discussion and Limitations

The central contribution is a modular geometric filtration layer that adaptively shapes retrieved context before a conventional reranker. The theory supplies an idealized separation mechanism, and the main experiments demonstrate useful end-to-end gains in selected QA regimes; the diagnostics below define the scope of that claim rather than replacing it.

This work focuses on Ricci-Filtration for QA tasks. However, applying it to sensemaking queries (query-focused summarization) is more complex (Edge et al., 2024), as these require analyzing global trajectories and entity connections. While performing discrete Ricci flow across the entire corpus could lead to GraphRAG-style community detection Edge et al. (2024), the computational cost of iterative curvature evaluation is currently prohibitive. Future research will focus on optimizing Ricci flow efficiency to support global sensemaking tasks on large-scale graphs.

Currently, we use cosine-dissimilarity thresholding to construct the initial edge-incidence graph for its simplicity. As shown in Table 3, the resulting filtration quality depends on the underlying embedding model. Exploring more robust graph construction remains a key research direction. On the other hand, experiments for HotpotQA and ablation study for MuSiQue (Table 6 in Appendix H) imply that current Ricci-Filtration has limitations in multi-hop reasoning where only critical and connected chunks should be used. How to combine the "geometric denoising" ability of discrete Ricci flow with the reasoning ability of RAG reranker appropriately is an important research direction.

The major limitation of Ricci-Filtration is its time complexity compared to other methods, as shown by Table 8 in Appendix K. While 20 iterations were used for most experiments, ablation studies suggest that 10 iterations remain effective. Consequently, developing a more time efficient iteration method with optimized weight cut-offs is also a critical direction for future application.

The matched controls sharpen the mechanism interpretation without changing the central contribution. The current flow score is a graph-structural context signal that precedes, rather than replaces, calibrated cosine or cross-encoder relevance; this division of labor explains how Ricci-Filtration can improve the complete pipeline even when a direct relevance score is stronger for standalone evidence ranking. Query conditioning currently enters mainly through topology, and the best topology is dataset dependent, making similarity-aware initialization a promising extension. The new diagnostics use one seed and 50 questions per dataset, whereas the larger original tables provide the primary end-to-end evidence; paired intervals and multiple random-filter draws would further quantify smaller effects. The online runtime study already covers 500 questions on one host, while provider-side OpenAI memory remains unobservable from the API client.

## Impact Statement

This paper presents a method for enhancing the performance of RAG with reranker through a differential geometry method. The broader impact of this work involves contributing to the development of more reliable and factually grounded AI assistants. This has direct applications in critical domains such as legal research, medical information retrieval, and education, where reducing "hallucinations" and ensuring source-verifiability are paramount for user safety and trust. On the other hand, while our method improves some evaluated settings, it remains susceptible to biases inherent in the underlying retrieval corpus. If the source data contains historical or social prejudices, the Ricci-Filtration may still prioritize these biased viewpoints, thereby amplifying them in the final generation. Furthermore, the deployment of multi-stage RAG pipelines increases the total inference-time computational overhead. We encourage practitioners to consider the environmental impact of increased energy consumption and to implement our system alongside robust bias-detection audits and data-governance frameworks to ensure equitable and sustainable deployment.

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

# Appendix

## A    Evaluation and audit protocols

### A.1    Comparison protocol and SQuADv2 scoring

All methods are evaluated from the same dense-retrieval pool and under the same downstream generation settings. For each query, the retriever returns the top 20 candidates using the same embedding model and cosine-similarity index. The no-reranker baseline sends the top five chunks directly to the generator, while the cross-encoder and LLM-based baselines rerank the top-20 pool and send five. Ricci-Filtration uses the same retrieval pool, generator, prompt, and metrics, with geometric filtering inserted before the same BGE reranker. The released Ricci pipeline sends all retained chunks, so its dynamic context is not matched to the five chunks used by the principal baselines. The comparison therefore measures the complete pipeline rather than isolating curvature from context amount.

For the released SQuADv2 experiment, we shuffle the validation split with seed 42, take the first 1,000 rows to construct the shared retrieval corpus, remove duplicate contexts, and split contexts into approximately 1,000-character chunks with 50-character overlap. The evaluation iterates over 1,000 questions from the same seeded stream. "Unanswerable" retains its official meaning relative to the original paragraph, but it is not guaranteed to remain unanswerable relative to every passage in this shared corpus. This result is therefore a shared-corpus stress test rather than an official single-paragraph reproduction. Exact evaluated identifiers and model snapshots were not recorded in the original table and must be logged in a confirmatory rerun.

Generated SQuADv2 answers are normalized by lowercasing, removing punctuation and articles, and collapsing whitespace. Token precision, recall, and F1 use token-multiset overlap and maximize over references. Examples without a reference span use the target "The question is unanswerable." The column labeled accuracy in Table 1 is not exact match: it is the fraction judged consistent by a temperature-zero GPT-4o-mini prompt. This LLM judge and the single run are additional sources of uncertainty.

### A.2    Completed SQuADv2 protocol pilot

We ran a balanced pilot of 100 answerable and 100 unanswerable validation questions with `gpt-4o-mini-2024-07-18`. In the source-paragraph-only protocol, normalized EM/F1 is 0.740/0.827 and no-answer accuracy is 0.840. With the same source paragraph anchored among 19 controlled distractors, EM/F1 is 0.680/0.761 and no-answer accuracy is 0.720. No prompt was truncated. The 0.066 F1 difference is driven mainly by unanswerable cases; answerable F1 is 0.814 versus 0.803. This pilot validates the protocol distinction but is not a full official-split evaluation.

### A.3    Audited pilot implementation

The supplementary pilots use an audited sequential POT implementation of the released package convention. The original end-to-end tables use the released GraphRicciCurvature path. Because version 0.5.3.2 hardcodes a Unix `fork` multiprocessing context that cannot run on the native Windows audit host, the new pilots use a sequential POT `emd2` implementation of the same convention: $\alpha = 0.5$, neighbor mass proportional to $\exp(-w^2)$, direct-edge curvature denominator, update–normalize–recompute order, and strict $w_{qj} < 1$ cutoff. A direct synthetic comparison against the package's internal curvature routine gives zero numerical difference; reproducing its tiny-edge shortcut changes audited query weights by less than $1.5 \times 10^{-8}$ and no retained set. These results are identified as independently reimplemented pilots rather than combined with the original package runtime.

## B    Query, topology, and trajectory diagnostics

Conditional on a fixed adjacency matrix, replacing the query embedding cannot affect a unit-initialized flow. All 120 frozen-topology swaps accordingly give retained-set Jaccard 1, decision-flip rate 0, and weight Spearman correlation 1. In the actual pipeline, however, the query embedding contributes to $\tau$ and adjacency.

Table 4: Approximately token-matched evidence retention, restricted to questions for which all annotated evidence is present in the initial 20-candidate pool. "Support recall" is the fraction of required evidence retained; "all evidence" is the fraction retaining every required item.

| Dataset | Eligible $n$ | Ricci support recall | Ricci all | Cosine all | BGE all | K-means all | Random all |
|---------|-----------|---------------------|-----------|------------|---------|-------------|------------|
| SQuADv2 | 25 | 0.760 | 0.760 | 1.000 | 1.000 | 1.000 | 1.000 |
| HotpotQA | 50 | 0.770 | 0.580 | 1.000 | 1.000 | 1.000 | 0.680 |
| MuSiQue | 50 | 0.812 | 0.580 | 0.900 | 0.760 | 0.800 | 0.560 |
| MultiHop-RAG | 35 | 0.762 | 0.429 | 0.743 | 0.800 | 0.743 | 0.486 |

Table 5: Normalized token F1 from the versioned matched GPT-4o-mini pilot. All datasets use a 40,000-token common cap and no selected document is truncated; all MultiHop-RAG methods tie at 0.680 EM.

| Dataset | Ricci | Cosine | BGE | K-means | Random |
|---------|-------|--------|-----|---------|--------|
| SQuADv2 | 0.628 | 0.616 | 0.657 | 0.637 | 0.628 |
| HotpotQA | 0.425 | 0.575 | 0.548 | 0.591 | 0.492 |
| MuSiQue | 0.214 | 0.163 | 0.203 | 0.154 | 0.136 |
| MultiHop-RAG | 0.681 | 0.681 | 0.697 | 0.680 | 0.680 |

When $\tau$ and topology are recomputed, paraphrase Jaccard is 0.978, 0.992, 0.934, and 0.873 on SQuADv2, HotpotQA, MuSiQue, and MultiHop-RAG; replacing the query by the least-similar diagnostic query gives 0.933, 0.798, 0.830, and 0.801. Initializing forced query edges with clipped cosine dissimilarity produces 0–1.5% additional decision flips under frozen-topology replacements. Thus query dependence in the released design enters principally through graph construction, while similarity-derived edge lengths offer a second conditioning channel for future variants.

No tested topology dominates. Similarity edges improve conditional all-evidence survival from 0.60 to 1.00 on HotpotQA and from 0.375 to 0.50 on MultiHop-RAG, but reduce it from 0.60 to 0.30 on MuSiQue and from 1.00 to 0.60 on the five answerable SQuADv2 cases. Symmetric $k$-NN and MST+$k$-NN are nearly identical because the $k$-NN graph is already connected in almost every case. In the corresponding GPT-4o-mini sensitivity test, similarity edges have the highest HotpotQA F1 (0.669), the current dissimilarity graph has the highest SQuADv2 (0.686) and MuSiQue (0.097) F1, and all four topologies tie at 0.700 on MultiHop-RAG. Native retained counts differ, so this is a design-sensitivity rather than matched-budget causal test.

## C   Matched-budget mechanism pilot

We freeze 50 examples per dataset with seed 42 and compare the 20-step Ricci ordering against deterministic random, dense-cosine, BGE-score, and $k = 2$ K-means orderings. Controls greedily use each query's Ricci-retained token budget and consume approximately 97.6–98.4% of it. SQuADv2 uses the source paragraph with controlled distractors, HotpotQA uses the hard-bridge context with padding, MuSiQue uses its benchmark paragraph set with padding, and only MultiHop-RAG uses corpus-wide retrieval. Random filtering uses one deterministic permutation.

Ricci retains 17.32, 16.88, 16.58, and 15.88 candidates on average, yet preserves all gold evidence less often than cosine or BGE on every dataset. Its evidence AUROCs are 0.485/0.310/0.508/0.430, versus 1.000/0.919/0.838/0.787 for cosine and 0.998/0.966/0.782/0.751 for BGE. For MultiHop-RAG, Table 4 conditions on the 35 of 38 evidence-bearing examples with complete initial coverage.

Every selected set is reordered by the same BGE reranker before `gpt-4o-mini-2024-07-18`. Ricci trails the best matched control on SQuADv2, HotpotQA, and MultiHop-RAG, while achieving the highest MuSiQue F1 (0.214 versus 0.203 for BGE). The 0.011 MuSiQue margin is promising but too small for a standalone superiority claim without a paired interval. On MultiHop-RAG, BGE wins two questions, Ricci wins two, and 46 tie in paired F1. This small matched pilot tests mechanism isolation under an imposed Ricci-sized

token budget; it complements the larger complete-pipeline results in Tables 1–2 rather than superseding them.

## D   Question Answering Dataset

For QA tasks, we use the following five widely used datasets:

1. **SQuAD2.0** Rajpurkar et al. (2018): The Stanford Question Answering Dataset (SQuAD) is a reading comprehension benchmark composed of questions created by crowdworkers based on a collection of Wikipedia articles. For each question, the answer is either a specific text span drawn from the associated passage or, in some cases, the passage does not contain an answer. SQuAD2.0 extends SQuAD1.1Rajpurkar et al. (2016) by retaining its original answerable questions and adding many adversarially written unanswerable questions that are designed to closely resemble answerable ones. To ensure a more challenging evaluation, we randomly selected 1,000 questions from the validation set of SQuAD2.0, which also involves unanswerable questions. Additionally, we treat SQuAD2.0 as a multi-document QA task and build a single RAG system to handle all questions, providing one test of whether Ricci-Filtration remains useful outside explicitly multi-hop benchmarks.

2. **TriviaQA** Joshi et al. (2017): A large-scale dataset with 650k+ question-answer-evidence triples. It is known for having more complex, "fact-heavy" questions that require a high-precision reranker to find the specific snippet of knowledge. Since the typical context is long for each question, we randomly selected 100 questions from the validation set of TriviaQA for evaluation and we build a uniform RAG system to handle all the selected questions.

3. **Hotpot**Yang et al. (2018) We utilize HotpotQA, a benchmark dataset for multi-hop reasoning that associates 10 context paragraphs with each question. Since modern LLMs can readily solve the dataset's simpler queries, we focus our evaluation on a more rigorous subset comprising 1,000 'hard bridging' questions randomly sampled from the validation set. We treat HotpotQA as a multi-document QA task, employing a unified RAG system to process all queries.

4. **MultiHop-RAG** Tang & Yang (2024) MultiHop-RAG is a QA dataset designed to evaluate retrieval and reasoning across multiple documents with metadata in RAG pipelines, where document metadata aims to reflect complex scenarios commonly found in real-world RAG applications. Constructed from English news articles, it contains 2,556 queries with supporting evidence distributed across 2 to 4 documents. The dataset includes four query types: Inference queries synthesize claims about a bridge entity to identify it; Comparison queries compare similarities or differences and yield "yes" or "no" answers in general; Temporal queries examine event ordering with answers like "before" or "after"; and Null queries where no answer can be derived from the retrieved documents. For evaluation, we treat it as a multi-document QA task.

5. **MuSiQue** Trivedi et al. (2021) MuSiQue (Multihop Single-hop Question Composition) is a challenging dataset for multi-hop Question Answering (QA) designed to be hard to cheat. It enforces connected reasoning, where one reasoning step critically relies on information from another. It was built bottom-up by first taking existing single-hop questions from datasets like SQuAD and Natural Questions, then systematically linking these questions together. Rigorous filters were employed to ensure the resulting multi-hop question could not be answered if any intermediate step was skipped. The dataset involves approximately 25,000 examples with 2-4 hop questions. Similarly, we randomly selected 1,000 questions from the validation set of MuSiQue and treated it as a multi-document QA task with a single RAG system handling all questions.

## E   Theoretical foundation

Note that most notations and definitions in this section are following the work in Ni et al. (2019). For more detailed introductions, interested readers can refer to the original paper of Ni et al. (2019).

### E.1 Notations and definitions

We represent a network as an unweighted graph $G = (V, E)$, defined by a vertex set $V$ and an edge set $E$. The objective of community detection is to partition $G$ into a collection of $n$ disjoint, connected subgraphs $\{C_1, C_2, \ldots, C_n\}$, referred to as communities. This is achieved by identifying a set of inter-community edges, those connecting distinct clusters, the removal of which isolates the subgraphs. Formally, a robust community structure is characterized by two fundamental properties

1. High Intra-community Density: Vertices within any given community $C_i$ exhibit a high degree of connectivity.

2. Low Inter-community Density: Vertices belonging to different communities are sparsely connected.

Such topological features are common in empirical networks, including social, biological, and technological systems

We will use the following definitions and conventions in Ni et al. (2019). Two vertices $i, j \in V$ are said to be adjacent ($i \sim j$) if there exists an edge $ij \in E$. We extend the unweighted model to a weighted graph (or metric graph) by defining a weight function $w : E \to \mathbb{R}_{\geq 0}$, which assigns a non-negative scalar $w_{ij}$ to each edge. A path of length $n$ between nodes $a$ and $b$ is defined as a sequence of edges $\{e_0, e_1, \ldots, e_{n-1}\}$ where $e_i = v_i v_{i+1}$, such that $v_0 = a$ and $v_n = b$. The length of the path $\{e_0, \ldots, e_{n-1}\}$ is defined to be $\sum_{i=0}^{n-1} w_{i(i+1)}$. The path is said to have n hops.

### E.2 The optimal transport problem

The classical formulation of the optimal transport problem, originally proposed by Gaspard Monge in 1781, concerns the identification of a transport map that minimizes the total cost associated with relocating mass from a source distribution to a target distribution. In its physical motivation, this corresponds to the efficient transfer of raw materials (e.g., iron ore) from production sites to industrial sinks. To begin, let us briefly recall the notion of metric spaces and Borel measures on a metric space. We define a metric space as a pair $(X; d)$ where $X$ is a set and $d$ is a distance function $d : X \times X \to \mathbb{R}_{>0}$ with the following properties:

- $d(x, y) = 0$ if and only if $x = y$;

- $d(x, y) = d(y, x)$;

- $d(x, y) + d(y, z) \geq d(x, z)$ for all $x, y, z \in X$.

On a given metric space $(X, d)$, we consider the Borel $\sigma$-algebra, generated by the collection of all open sets in $X$ under countable unions, countable intersections, and relative complements. A Borel probability measure $\mu$ is a mapping from the Borel sets to the interval $[0, 1]$ that satisfies:

- Normalization: $\mu(X) = 1$

- Countable Additivity: For any sequence of pairwise disjoint Borel sets $\{A_i\}_{i=1}^{\infty}$,

$$\mu \left( \bigcup_{i=1}^{\infty} A_i \right) = \sum_{i=1}^{\infty} \mu(A_i)$$

In the context of a finite graph $G = (V; E)$, the metric space $(X, d)$ is specialized by setting $X$ to be the discrete vertex set $V$. Under the discrete topology, the Borel $\sigma$-algebra is equivalent to the power set $\mathcal{P}(V)$; thus, every subset of $V$ is a Borel set. A Borel probability measure $\mu$ on $V$ is consequently characterized by a probability mass function $\mu : V \to [0, 1]$ that satisfies the normalization condition $\sum_{i \in V} \mu(i) = 1$.

Given an edge weighted graph $(V; E; w)$ with $w_{ij} > 0$ for all edges, one introduces a metric $d$ on vertex set $V$ by

$$d(v, v') = \min_{\gamma: v \rightsquigarrow v'} \sum_{e \in \gamma} w_e. \tag{9}$$

where the minimum is taken over all edge paths from $v$ to $v'$. We call $d$ the induced metric from the edge weight $w$. Note that by definition, $d(v, v') + d(v', v'') \geq d(v, v'')$ for any three vertices $v, v', v'' \in V$. This is exactly the definition we give in section 2.2.

Consider two metric spaces, $X$ and $Y$, endowed with Borel probability measures $\mu$ and $\nu$, respectively. In the physical context of the Monge problem, $X$ and $Y$ represent the spatial distributions of supply centers (e.g., mines) and demand centers (e.g., factories), while $\mu$ and $\nu$ characterize the respective densities of material to be transported and consumed. We define a continuous cost function $c : X \times Y \to \mathbb{R}_{>0}$, where $c(x, y)$ denotes the cost incurred by transporting a unit of mass from location $x \in X$ to $y \in Y$. In the specific case where $X = Y$ and the transportation cost is proportional to the displacement under a constant rate, the cost function is typically identified with the metric of the underlying space, such that $c(x, y) = d(x, y)$

A transport map $T : (X, \mu) \to (Y, \nu)$ is a measure preserving map, i.e., for any Borel set $A \subset Y$, $\nu(A) = \mu(T^{-1}(A))$. Monge's formulation of the optimal transportation problem is to find a transport map $T : X \to Y$ that realizes the infimum

$$\inf \left\{ \int_X c(x, T(x)) d\mu(x) \middle| \text{T is a transportation} \right\}$$

A transformation $T : X \to Y$ that achieves the aforementioned infimum is termed an optimal transport map. In this general setting, the existence of such a map is not guaranteed by classical results, as the Monge formulation is highly non-linear and subject to strict feasibility constraints. A significant advancement in the field was established by Kantorovitch (1958), who reformulated the problem into a linear optimization framework, ensuring the existence of a solution. Kantorovich relaxed the deterministic map $T$ into a transportation plan $\gamma$, defined as a Borel probability measure on the product space $X \times Y$. To be considered a valid coupling, $\gamma$ must satisfy $\gamma(A \times Y) = \mu(A)$ and $\gamma(X \times B) = \mu(B)$ for all Borel sets $A$ and $B$. The goal is to find a transportation plan $\gamma$ that attains the infimum cost

$$W(\mu, \nu) = \inf \left\{ \int_{X \times Y} c(x, y) d\gamma(x, y) \middle| \gamma \in \Gamma(\mu, \nu) \right\}$$

where $\Gamma(\mu, \nu)$ denotes the collection of all transportation plans. If $X = Y$, the quantity $W(\mu, \nu)$ is called the Wasserstein distance between two probability measures $\mu, \nu$ on $X$. Kantorovich proved that the infimum is always achieved by some transportation plan.

In the context of a finite weighted graph $G = (V, E, w)$, the Kantorovich problem admits a discrete reformulation. Let $d : V \times V \to \mathbb{R}_{\geq 0}$ denote the path metric induced by the edge weights, as defined in Equation (1). A transportation plan $\gamma$ is given by a map $\gamma : V \times V \to [0, 1]$ such that $\sum_{i \in V} \gamma_{ij} = \mu_j$ and $\sum_{j \in V} \gamma_{ij} = \nu_i$ for all $i, j \in V$. The goal is to find the minimum cost

$$min \left\{ \sum_{i,j \in V} \gamma_{i,j} d(i, j) : \gamma \text{ is a transportation plan} \right\}$$

which is a linear programming problem and solution is computationally tractable using standard optimization algorithms, such as the simplex method or interior-point methods, making Wasserstein distance a robust metric for analyzing distances between distributions on graphs.

### E.3 Curvatures in classical differential geometry

A foundational pillar of contemporary geometry is the concept of curvature, a quantitative measure of spatial deviation from flatness. Formally conceptualized by Gauss and Riemann in the 19th century, the study of curvature begins with the $n$-dimensional manifold: a topological space that is locally homeomorphic to

Euclidean $n$-space. When such a manifold is equipped with a Riemannian metric, which assigns an inner product to each tangent space, it becomes a Riemannian manifold, the primary object of inquiry in the field.

Historically, these concepts emerged from the study of smooth surfaces ($n = 2$) embedded in $\mathbb{R}^3$. For a surface $S$, the Gauss map $\mathcal{G} : S \to S^2$ maps each point $p \in S$ to its unit normal vector. The Gaussian curvature $K$ at $p$ is defined as the Jacobian determinant of this map, representing the signed area distortion.

Under this framework, geometric topographies are categorized by their curvature: Euclidean planes exhibit zero curvature ($K = 0$), spheres exhibit constant positive curvature ($K > 0$), and hyperboloids of one sheet exhibit negative curvature ($K < 0$). A pivotal advancement was Gauss's Theorema Egregium, which demonstrated that Gaussian curvature is an intrinsic property. It depends solely on the Riemannian metric and remains invariant regardless of how the surface is isometrically embedded in higher-dimensional space.

Riemann extended these principles to higher dimensions through the introduction of sectional curvature. For a Riemannian manifold $(M, g)$, the sectional curvature $K(P)$ assigns a scalar value to each 2-dimensional plane $P$ within the tangent space $T_pM$. This scalar corresponds to the Gaussian curvature of the surface formed by the image of $P$ under the exponential map.

The sign of the curvature dictates the global behavior and "density" of the manifold. Positively curved spaces tend toward "crowded" geometries with small diameters, whereas negatively curved spaces exhibit a "spreading" geometry, often characterized by infinite fundamental groups, contractible universal covers, and large-scale branching similar to a tree structure.

The Ricci curvature is a fundamental tensor that assigns a scalar value to each unit tangent vector $v$ at a point $p$, representing the average of the sectional curvatures of all 2-dimensional planes containing $v$. In modern geometry, this curvature is often characterized by its influence on the metric properties of the space. Specifically, Ricci curvature governs the volume growth rate of geodesic balls relative to their radii. Furthermore, it determines the volume of intersection between two overlapping balls as a function of their radii and the displacement between their respective centers.

There have been various approaches to generalize the notion of curvature to spaces which are not manifolds (e.g., a graph with edge weights). One of the important work of Ollivier (2007) which relates Ricci curvature to optimal transport. Since optimal transport can be formulated on very general metric spaces with probability measure at each point, in particular on networks with edge weights and probability measures at each vertex, it facilitates the application in community detection using curvature and optimal transport (Ni et al., 2019).

### E.4   The Ricci flow

Introduced by Hamilton (1982), the Ricci flow is a fundamental tool in geometric analysis that deforms the metric of a Riemannian manifold through a process formally analogous to the diffusion of heat. By evolving the metric in a way that smooths out geometric irregularities, the flow serves as a non-linear counterpart to the heat equation. Over the past forty years, it has emerged as one of the most powerful techniques for resolving profound geometric problems, most notably providing the framework for the proof of the Poincare conjecture.

Given a Riemannian manifold $M$ equipped with a metric $g_{ij}$ and its corresponding Ricci curvature tensor $R_{ij}$, Hamilton's Ricci flow is defined as a second-order nonlinear partial differential equation governing the evolution of symmetric $(0, 2)$-tensors:

$$\frac{\partial}{\partial t} g_{ij}(t) = -2R_{ij}(t)$$

A solution to the Ricci flow is a one-parameter family of metrics $g_{ij}(t)$ on a smooth manifold $M$ satisfying the above partial differential equation. One of the key properties of the Ricci flow is that the curvature evolves according to a nonlinear version of the heat equation. Thus the Ricci flow tends to smooth out irregularity of the curvature. Under the Ricci flow, regions in the manifold of positive sectional curvature tend to shrink and regions of negative sectional curvature tend to expand and spread out. Singularities usually occur while deforming a Riemannian 3-manifold through the Ricci flow. They appear in a small

neighborhood of a surface in the 3-manifold. By removing the singularities (i.e., surfaces) and redefining the Ricci flow on the remaining pieces, one produces the Ricci flow with surgery on the manifold. The ground-breaking work of Perelman (2002) shows that the Ricci flow with surgery captures the geometric decomposition of the 3-manifold. It solves the Geometrization Conjecture of Thurston and geometrically classifies all 3-manifolds.

Ricci flow has been used as a framework for analyzing network evolution and community detection. Under this heuristic, a network is treated as a discrete approximation of a high-dimensional manifold (such as a 3-manifold), where distinct network communities correspond to the components of a geometric decomposition. Since Perelman (2002) used Ricci flow to identify geometric components of 3-manifolds, related discrete Ricci-flow methods have been explored for uncovering community structures within networks. Analogous to the Hamilton-Perelman framework, the specific iteration cutoffs and surgery thresholds for the flow must be calibrated to the unique topological properties of each individual network.

## F  Proof of Theorem 3.1

We start by computing the Wasserstein distance of a metric on $G(a; b)$. Note that in each community $C_i$ there is a specific node $u_i$ which connects to other communities. According to the proof in Ni et al. (2019), we call this node the gateway node and the rest of the nodes in $C_i$ the non-gateway nodes. As shown in figure 4, there are three types of edges in the graph:

1. Edges connecting two communities (on two gateway nodes, such as $uv$ in figure 4;

2. Edges connecting a gateway node with a non-gateway node in the same community, such as $ui, uj$ figure 4;

3. And edges connecting two non-gateway nodes (such as $ij$).

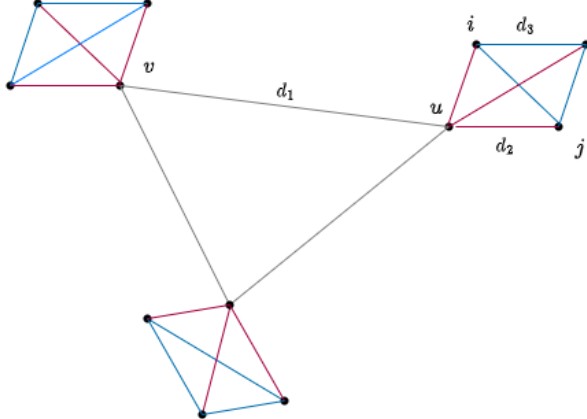

Figure 4: An example graph $G(a; b)$ obtained from a complete graph on $b + 1$ vertices by replacing each vertex by a complete graph of $a + 1$ vertices. In this figure, $a = 3, b = 2$. This plot is essentially the same as the one given by Ni et al. (2019)

In Section 3.1, the initial metric assigns unit length to each edge of the constructed graph. In addition, iteration (5) implies that the Ricci flow preserves the graph symmetry. Note that there are only three different edge lengths at each iteration of the Ricci flow, corresponding to the three types of edges. In addition, we assume the edge lengths of the $(n+1)$-th iteration be $W_1$, $W_2$ and $W_3$ which are the Wasserstein distances of the corresponding edges for the metric graph $(G(a; b); d)$ with respect to the probability measures $\{\mu_x | x \in V\}$.

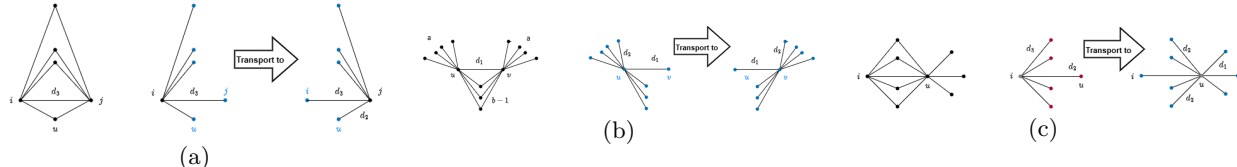

Figure 5: Parts (a), (b) and (c) from Ni et al. (2019) illustrate the optimal transportation to move the mass at vertex $u$ to vertex $v$ under different types of edges.

Suppose the edge lengths of the metric at the $n$-th iteration are $d_{n,1}$, $d_{n,2}$ and $d_{n,3}$ for the edges between communities, edges from a gateway node to a non-gateway node, and edges between two non-gateway nodes respectively, as shown in figure 4. Then we have $d_{n,1} = w_{uv} = W_{n,1}, d_2 = w_{uj} = W_{n,2}, d_3 = w_{ij} = W_{n,3}$ since there is only one path connecting each pair. This observation is critical in the following derivation. Lemma F.1 facilitates the asymptotic analysis of edge lengths under normalized discrete Ricci flow in Algorithm 1. For light notation, we ignore the number of iteration subscript $(n, n+1)$ for Wasserstein distance and edge length in the proof of Lemma F.1.

**Lemma F.1.** *According to the notations above and assumptions in Theorem statement, the Wasserstein distances $W_1, W_2, W_3$ are given by*

- $W_1 = \frac{a-1}{a+b}d_1 + \frac{2a}{a+b}d_2$

- $W_2 = \frac{b}{a+b}d_1 + \frac{ab-a-b}{a(a+b)}d_2$

- $W_3 = \frac{1}{a}d_3$

*In addition, assume that $a > b \geq 2$ and $d_1 \geq d_2 \geq d_3$. Then we have $W_1 \geq W_2 \geq W_3$.*

*Proof.* We derive the identity of $W_3$ first as it is the most straightforward case shown in Figure 5(a). Note that nodes $i$ and $j$ share the same vertical vertices. Thus we only need to move mass at node $i$ $(1/a)$ to node $j$ along $ij$ to finish the transport, which is optimal by definition. It follows that $W_3 = \frac{1}{a}d_3$.

Next, we consider the identity of $W_1$, which moves the probability measure $\mu_u$ to $\mu_v$. By definition and Figure 5(b), the degrees of nodes $u$ and $v$ are all equal to $a + b$. By the node distribution parameter, both probability measures $\mu_u$ and $\mu_v$ have density(mass) $\frac{1}{a+b}$ at each vertices connected to $u$ and $v$. Note that $u$ and $v$ share the middle $b - 1$ vertices where each of them has mass $\frac{1}{a+b}$ under both measures $\mu_u$ and $\mu_v$. For the sake of minimal effort in transportation from $\mu_u$ to $\mu_v$, there is no need to move them. For the rest of $a$ many vertices $x$ adjacent to $u$ and $a$ many vertices $y$ adjacent to $v$, we consider the following transportation plan, which is the best one:

1. For each vertex $x$ adjacent to $u$, where the mass at $x$ is $\frac{1}{a+b}$, we move it along edge $xu$ from $x$ to $u$. The total cost of moving all of them is $\frac{a}{a+b}d_2$.

2. Leave a mass of $\frac{1}{a+b}$ at $u$ and move the rest of mass to $v$, where the total cost would be $\frac{a-1}{a+b}d_1$.

3. After the previous two steps, the total mass at vertex $v$ is $\frac{a}{a+b} = \frac{a-1}{a+b} + \frac{1}{a+b}$. We now evenly distribute those mass to the $a$ many vertices $y$ adjacent to $v$, where the total cost would be $\frac{a}{a+b}d_2$

Therefore, $W_1 = \frac{a-1}{a+b}d_1 + \frac{2a}{a+b}d_2$.

As to the equation for $W_2$, notice that the mass of $\mu_i$ at a vertex $x$ adjacent to $i$ is $\frac{1}{a}$ and the mass of $\mu_u$ at a vertices adjacent to $u$ is $\frac{1}{a+b}$. In addition, every vertex adjacent to $i$ is also adjacent to $u$. The optimal transportation is as follows:

1. Leave the mass $\frac{1}{a+b}$ at each vertex $x \neq u$ adjacent to $i$. Then the total leftover mass at those vertices is $(a-1)(\frac{1}{a} - \frac{1}{a+b}) = \frac{(a-1)b}{a(a+b)}$. Move the total mass of $\frac{1}{a+b}$ from the leftover mass at these $x$ to the vertex $i$ of distance $d_3$ from $x$. Note that there are many ways to achieve this. For example, for each of these $(a-1)$ vertices, we move $\frac{1}{(a+b)(a-1)}$ from each of them to $i$, then the total cost would be $\frac{d_3}{a+b}$. This step finishes the mass transportation for $i$.

2. After step 1, there is a total mass of $\frac{(a-1)b}{a(a+b)} - \frac{1}{a+b} = \frac{ab-a-b}{a(a+b)}$ at these x. Similarly, we can move the mass to vertex $u$ along edges of length $d_2$, generating total cost $\frac{ab-a-b}{a(a+b)}d_2$.

3. Now the mass at vertex $u$ becomes $\frac{1}{a} + \frac{ab-a-b}{a(a+b)} = \frac{b}{a+b}$. We then evenly distribute this mass to vertices $y$ adjacent to $u$ such that $y$ is not adjacent to $i$. The total cost is $\frac{bd_1}{a+b}$.

Therefore, $W_2 = \frac{d_3}{a+b} + \frac{ab-a-b}{a(a+b)}d_2 + \frac{b}{a+b}d_1$.

To show the inequality $W_1 \geq W_2 \geq W_3$, we first have $W_2 \geq \frac{b}{a+b}d_1 \geq \frac{1}{a}d_1 \geq \frac{1}{a}d_3 = W_3$. On the other hand,

$$
\begin{aligned}
W_1 - W_2 &= \frac{a-1}{a+b}d_1 + \frac{2a}{a+b}d_2 - \left( \frac{d_3}{a+b} + \frac{ab-a-b}{a(a+b)}d_2 + \frac{b}{a+b}d_1 \right) \\
&= \frac{a-1-b}{a+b}d_1 + \frac{2a^2 - ab + a + b}{a(a+b)}d_2 - \frac{1}{a+b}d_3 \\
&= \frac{a(a-1-b)}{a(a+b)}d_1 + \frac{2a^2 - ab + a + b}{a(a+b)}d_2 - \frac{a}{a(a+b)}d_3 \\
&\geq \frac{d_2}{a(a+b)}\left[ a(a-b-1) + 2a^2 - ab + a + b - a \right] \\
&=\geq \frac{d_2}{a(a+b)}\left[ a(a-b-1) + a^2 - ab + a^2 + b \right] \geq 0 \quad \text{since } a > b \geq 2 \text{ by assumption}
\end{aligned}
$$
(10)

$\square$

Now we begin the proof of Theorem 3.1.

*Proof.* Consider a $3 \times 3$ matrix $A$ such that

$$
A = \begin{bmatrix} \frac{a-1}{a+b} & \frac{2a}{a+b} & 0 \\ \frac{b}{a+b} & \frac{ab-a-b}{a(a+b)} & \frac{1}{a+b} \\ 0 & 0 & \frac{1}{a} \end{bmatrix}
$$

For the theoretical recurrence below, we use the unit-step normalized update, i.e., the case $\varepsilon = 1$ in (5). Since $\kappa = 1 - W/d$, this sends each edge length to the corresponding Wasserstein distance before normalization. By Lemma F.1 and the normalization constraint in Algorithm 1, we have the following constrained system of difference equations:

$$
\begin{bmatrix} w_{n+1,1} \\ w_{n+1,2} \\ w_{n+1,3} \end{bmatrix} = A \begin{bmatrix} w_{n,1} \\ w_{n,2} \\ w_{n,3} \end{bmatrix}, \quad w_{n,1} + w_{n,2} + w_{n,3} = |E|
$$
(11)

where $|E|$ is the total number of edges for the given graph. We can rewrite the system of difference equations above as $W_{n+1} = AW_n$ by denoting $W_n = [w_{n,1}, w_{n,2}, w_{n,3}]^T$. Here $w_{n,i}$ represents the length of the $i$-th edge type in the graph $G(a;b)$ described in Figure 5 after the $n$-th iteration of the Ricci flow. We use the observation that $d_1 = w_{n,1}$, $d_2 = w_{n,2}$, and $d_3 = w_{n,3}$.

We then introduce another lemma given by Ni et al. (2019)

**Lemma F.2.** *Suppose $a > b \geq 2$, there are three real eigenvalues satisfying $\lambda_1 > \lambda_2 = \frac{1}{a} \geq 0 > \lambda_3$. In addition, one eigenvector $e_1$ associated to $\lambda_1$ is of the form $[1, k, 0]^T$ where $0 < k < 1$.*

Suppose the eigenvectors associated to $\lambda_2$ and $\lambda_3$ are $e_2$ and $e_3$ respectively, it's easy to see $e_2 = [0, 0, 1]^T$ for $\lambda_2 = \frac{1}{a}$. On the other hand, all eigenvalues are distinct, it follows that $e_1, e_2, e_3$ are linearly independent in $\mathbb{R}^3$. Under the settings in Algorithm 1, the initial condition becomes $w_{0,1} = w_{0,2} = w_{0,3} = 1$.

Note that the general solution for system (11) can now be written as

$$
W_n = c_1 \lambda_1^n e_1 + c_2 \lambda_2^n e_2 + c_3 \lambda_3^n e_3 = \begin{bmatrix} c_1 \lambda_1^n + o(\lambda_1^n) \\ k c_1 \lambda_1^n + o(\lambda_1^n) \\ \left(\frac{1}{a}\right)^n \end{bmatrix}
$$

where $c_1, c_2, c_3$ can be solved by initial condition and constraint condition. We only keep $c_1$ to simplify the expression, which will not affect the conclusion. The small o notation $o(\lambda_1^n)$ represents a function $f(n) = o(\lambda_1^n)$ that grows strictly slower than $o(\lambda_1^n)$ as $n$ approaches infinity, i.e. $\lim_{n \to \infty} o(\lambda_1^n)/\lambda_1^n = 0$.

As a conclusion, we see that the length of edge type $uv$ ($w_{n,1}$) grows at the rate of $\lambda_1^n$, the length of edge type $ui$ ($w_{n,2}$) and $ij$ ($w_{n,3}$) grows at rate $o(\lambda_1^n)$. More specifically, the length of edge type $ij$ ($w_{n,3}$) shrinks at the rate of $\frac{1}{a} < 1$ and shrinks to zero exponentially fast. Namely, the weight of the intra-community edges shrink asymptotically faster than the weight of the inter-community edges.

$\square$

# G   Finite-time extension for $\alpha = 1/2, p = 2$

The proof of Theorem 3.1 above treats the uniform-neighborhood case $\alpha = 0, p = 0$, where the Wasserstein update reduces to a linear recurrence on the three edge types. The parameter choice used in our experiments, $\alpha = 1/2, p = 2$, is different: the probability measure at a node depends on the current edge lengths through the exponential term in (2). Therefore the same eigenvalue argument does not directly apply. We prove Proposition 3.2 by using direct estimates on the nonlinear Wasserstein update.

*Proof of Proposition 3.2.* Under the ordering $d_1 \geq d_2 \geq d_3 > 0$, the direct edge realizes the shortest path between the endpoints of each edge type. Thus the neighbor distances entering (2) are exactly the corresponding edge-type lengths $d_1, d_2, d_3$. Write

$$
E_\ell = \exp(-d_\ell^2), \qquad C_g = a E_2 + b E_1, \qquad C_n = E_2 + (a-1)E_3.
$$

For a gateway node $u$, the mass distribution has mass $1/2$ at $u$, mass $E_2/(2C_g)$ at each of the $a$ non-gateway neighbors in the same community, and mass $E_1/(2C_g)$ at each of the $b$ gateway neighbors in other communities. For a non-gateway node $i$, the distribution has mass $1/2$ at $i$, mass $E_2/(2C_n)$ at its gateway node, and mass $E_3/(2C_n)$ at each of the other $a - 1$ non-gateway nodes in the same community.

By the same symmetry and shortest-path transport argument used in Lemma F.1, the three Wasserstein distances are

$$
W_1 = \frac{2a E_2 + (b-1) E_1}{2C_g} d_1 + \frac{a E_2}{C_g} d_2, \tag{12}
$$

$$
W_2 = \frac{(a-1) E_3}{2C_n} d_2 + \frac{b E_1}{2C_g}(d_1 + d_2), \tag{13}
$$

$$
W_3 = \frac{E_2 + (a-2) E_3}{2C_n} d_3. \tag{14}
$$

For example, in the type-3 case the common mass at the gateway and at the other shared non-gateway neighbors cancels, and only the excess mass at one endpoint must be transported across the edge to the

other endpoint. The type-1 and type-2 formulas follow by canceling common mass and then moving the remaining surplus along shortest paths; each displayed plan attains the matching lower bound obtained by counting the mass that must cross the corresponding gateway and within-community edge classes.

Let $X = E_1/E_2$ and $Y = E_3/E_2$. Since $d_1 \geq d_2 \geq d_3$, we have $0 < X \leq 1$ and $Y \geq 1$. From (13) and (14),

$$W_2 - W_3 = \frac{(a-1)Y d_2 - \big(1 + (a-2)Y\big)d_3}{2(1 + (a-1)Y)} + \frac{bX}{2(a+bX)}(d_1 + d_2).$$

The first term is nonnegative because $d_2 \geq d_3$ and $Y \geq 1$, and the second term is strictly positive. Thus $W_2 > W_3$.

Similarly, (12) and (13) give

$$W_1 - W_2 = \frac{2a - X}{2(a+bX)}d_1 + \left[\frac{2a - bX}{2(a+bX)} - \frac{(a-1)Y}{2(1 + (a-1)Y)}\right]d_2.$$

If the bracketed coefficient is negative, the assumption $d_1 \geq d_2$ yields the lower bound

$$W_1 - W_2 \geq \left[\frac{4a - (b+1)X}{2(a+bX)} - \frac{(a-1)Y}{2(1 + (a-1)Y)}\right]d_2.$$

The first fraction is minimized at $X = 1$, while the second fraction is strictly smaller than $1/2$. Hence

$$W_1 - W_2 > \left[\frac{4a - b - 1}{2(a+b)} - \frac{1}{2}\right]d_2 = \frac{3a - 2b - 1}{2(a+b)}d_2 > 0,$$

where the last inequality follows from $a > b \geq 2$. Therefore $W_1 > W_2$.

Finally, (14) implies

$$W_3 \leq \frac{a-1}{2a}d_3,$$

because the coefficient of $d_3$ is maximized when $Y = 1$. Also (12) implies

$$W_1 \geq \frac{2a + b - 1}{2(a+b)}d_1,$$

because the coefficient of $d_1$ is minimized when $X = 1$. Combining the two estimates gives

$$\frac{W_3}{W_1} \leq \frac{(a-1)(a+b)}{a(2a+b-1)}\frac{d_3}{d_1} = q_{a,b}\frac{d_3}{d_1}.$$

Since $a > b \geq 2$, $q_{a,b} < 1$. The normalized Ricci-flow step multiplies all updated edge lengths by the same positive normalization factor, so it preserves both ordering and ratios. Starting from $d_1 = d_2 = d_3 = 1$, induction gives the claimed ordering preservation and geometric decay of $w_{n,3}/w_{n,1}$. $\qquad\square$

*Remark* G.1. Proposition 3.2 is intentionally weaker than Theorem 3.1. It handles the practical lazy distribution $\alpha = 1/2, p = 2$, but only proves ordering preservation and relative contraction of the non-gateway intra-community edge under the symmetric $G(a,b)$ model and the unit-step normalized update. It should not be read as a full convergence theorem for arbitrary embedding-derived graphs or for every finite-step implementation detail of Algorithm 1.

## H Ablation study for MuSiQue

Table 6: Ablation studies of Ricci-Filtration on the Accuracy (%) of MuSiQue dataset

| Methods | $k$ | $n^*$ | $\tau$ | $p$ | $M$ | Embedding model | Reranker model | Acc |
|---|---|---|---|---|---|---|---|---|
| Cross-Encoder | 20 | 5 | - | - | - | text-embedding-3-small | bge-reranker-base | **29.30** |
| Cross-Encoder | 20 | 10 | - | - | - | text-embedding-3-small | bge-reranker-base | **30.50** |
| Cross-Encoder | 20 | 5 | - | - | - | text-embedding-ada-002 | bge-reranker-base | **30.20** |
| Cross-Encoder | 20 | 5 | - | - | - | text-embedding-3-small | ms-marco-MiniLM-L6-v2 | **28.90** |
| Ricci-Filtration | 10 | 6.5 | 50% | 2 | 20 | text-embedding-3-small | bge-reranker-base | 28.10 |
| Ricci-Filtration | 20 | 15.7 | 50% | 2 | 20 | text-embedding-3-small | bge-reranker-base | 28.80 |
| Ricci-Filtration | 20 | 15.3 | 50% | 5 | 20 | text-embedding-3-small | bge-reranker-base | 28.50 |
| Ricci-Filtration | 20 | 15.3 | 50% | 1 | 20 | text-embedding-3-small | bge-reranker-base | 27.10 |
| Ricci-Filtration | 20 | 16.6 | 25% | 2 | 20 | text-embedding-3-small | bge-reranker-base | **29.20** |
| Ricci-Filtration | 20 | 14.4 | 75% | 2 | 20 | text-embedding-3-small | bge-reranker-base | 26.20 |
| Ricci-Filtration | 20 | 14.6 | 50% | 2 | 10 | text-embedding-3-small | bge-reranker-base | 28.30 |
| Ricci-Filtration | 20 | 10.2 | 50% | 2 | 1 | text-embedding-3-small | bge-reranker-base | 23.20 |
| Ricci-Filtration | 20 | 16.1 | 50% | 2 | 20 | text-embedding-ada-002 | bge-reranker-base | 26.40 |
| Ricci-Filtration | 20 | 16.4 | 50% | 2 | 20 | text-embedding-3-small | ms-marco-MiniLM-L6-v2 | 28.60 |

*Note.* The better performing values are highlighted in bold. The results from different reranker models are highlighted with underline. $n$ for Ricci-Filtration methods represents the average number of chunks fed into reranker model after Ricci-Filtration.

## I K-means filtering ablation

To separate the effect of Ricci-flow-based graph filtration from a simpler clustering heuristic, we compare Ricci-Filtration with a K-means filtering baseline. The K-means baseline clusters the query and initially retrieved chunk embeddings into two clusters ($k = 2$), keeps the cluster containing the query node, and then follows the same downstream RAG procedure. For Ricci-Filtration, we use $\alpha = 0.5$ and $p = 2$; both methods use the same initial retrieval setting and gpt-4o-mini generation setting.

Table 7: Accuracy (%) comparison between Ricci-Filtration and K-means filtering with $k = 2$.

| Methods | SQuADv2 | HotpotQA | MultiHop Inference | MultiHop Comparison | MultiHop Temporal | MultiHop Null |
|---|---|---|---|---|---|---|
| Ricci-Filtration | **73.00** | 67.00 | **97.18** | 64.02 | **56.60** | **95.00** |
| K-means filtering ($k = 2$) | 63.70 | **67.70** | 88.48 | **67.99** | 50.77 | 94.02 |

The comparison is mixed but informative. Ricci-Filtration improves four of the six reported accuracy metrics, including SQuADv2 and three MultiHop-RAG query types. K-means filtering is slightly better on HotpotQA and MultiHop comparison queries. Because K-means and Ricci-Filtration are not matched per query for retained chunk count or token budget, this table is appropriately interpreted as an end-to-end comparison of two adaptive pipelines. The approximately token-matched pilot in Section C answers the stricter isolation question: cosine, BGE, and usually K-means preserve more annotated evidence, whereas Ricci-Filtration attains the highest matched-generation F1 on MuSiQue. Together, the two experiments support the main conclusion that Ricci flow is a useful adaptive context signal in selected regimes without requiring it to dominate every ordinary pruning rule.

## J  LLM-based reranker prompt

For the LLM-based reranker baseline, each query is paired with the initially retrieved passages, formatted as `Document i:  ....` The model is instructed to return a JSON dictionary mapping each passage id to a relevance score from 0 to 10. The system prompt and user-message template are shown below.

### System prompt.

```
You are a customer support answer service. Your task is to evaluate help
center passages and score their relevance to a given customer query for a
retrieval augmented generation (RAG) system.

Evaluation Process:
1. Analyze the customer's query to identify both explicit needs and implicit
   context, including underlying user goals.
2. Assess each passage's ability to directly resolve the query or provide
   substantive supporting information with actionable guidance.
3. Score based on how effectively the passage addresses the query's core
   intent while considering potential interpretations.

Grading Criteria:
<grading_scale>
10: EXCEPTIONAL match - Contains exact step-by-step instructions that
    perfectly match the query's specific scenario. Must include all required
    parameters/context and resolve the issue completely without any ambiguity.
    Reserved for definitive solutions that exactly mirror the user's
    described situation and require no interpretation.

9: NEAR-PERFECT solution - Contains all critical steps for resolution but may
   lack one minor non-essential detail. Addresses the precise query parameters
   with specialized information. Solution must be directly applicable without
   requiring adaptation or assumptions.

8: STRONG MATCH - Provides complete technical resolution through specific
   instructions, but may require simple logical inferences for full
   application. Covers all essential components but might need minor
   contextualization.

7: GOOD MATCH - Contains substantial relevant details that address core
   aspects of the query, but lacks one important element for complete
   resolution. Provides concrete guidance requiring some user interpretation.

6: PARTIAL match - General guidance on the right topic but lacks the specifics
   for direct application. May only resolve a subset of the request.

5: LIMITED relevance - Related context or approach, but indirect. Requires
   substantial effort to adapt to the user's exact need.

4: TANGENTIAL - Mentions related concepts/keywords with little practical
   connection to the request. Minimal actionable value.

3: VAGUE domain info - Talks about the general area but not the query's
   specifics. No concrete, actionable steps.

2: TOKEN overlap - Shares isolated terms without context or intent aligned to
   the request. Similarity is coincidental.

1: IRRELEVANT - Uses query terms in a completely unrelated way. No meaningful
   link to the user's goal.

0: UNRELATED - No thematic or contextual connection to the query at all.
</grading_scale>
```

### User message template.

```
Question:
{query}
```

```
Context:
Document 0: {passage_0}

Document 1: {passage_1}

...

Output Format:
<output_format>
Return your response in a valid JSON (skip spaces):
{"id0":score0,"id1":score1,...}

Strict guidelines:
- Return ONLY a well-formed valid JSON with passage IDs as keys
- Each key must be a passage id in the format "idN"
- Integer values only, no decimals
- Skip spaces in the JSON
- No additional text or formatting
- Maintain original passage ID order
</output_format>

Reranked documents:
```

## K  Time efficiency comparison

Table 8: Average and standard deviation running time (in seconds) for each QA task under different methods on the current ten-query pilot.

| METHODS | SQUADv2 | HOTPOTQA | MUSIQUE |
|---|---|---|---|
| CROSS-ENCODER RERANKER | 1.08± 0.22 | 1.37 ± 0.30 | 4.18 ± 0.88 |
| LLM AS RERANKER | 3.75± 0.22 | 4.82± 0.95 | 4.25 ± 1.14 |
| RICCI-FILTRATION | 8.62± 0.32 | 9.41 ± 0.47 | 12.37 ± 1.21 |

The legacy end-to-end timing in Table 8 uses only ten questions and a different pipeline, so it should not be numerically combined with the new measurements. The flow-only study in Table 9 runs one process with `OMP_NUM_THREADS=1` on an Intel Core i7-12700H under Windows and Python 3.12.7. From $k = 10$ to $k = 40$, the mean number of graph edges grows from 28–34 to 417–437, the number of exact optimal-transport solves grows from 588–710 to 8,757–9,182, and mean time grows by roughly 14–20×. Every run reaches the 20-step cap. At $(k, M) = (80, 20)$ in the supplementary stratified matrix, mean flow-only time is 8.89–9.38 seconds, with about 1,634–1,670 edges and 34,316–35,064 OT solves.

Table 10 closes the client-visible deployment gap using an NVIDIA RTX 4070 Laptop GPU and the same CPU. For 500 frozen questions from each of four datasets, both paths make uncached live `text-embedding-3-small` and `gpt-4o-mini-2024-07-18` calls, search the same precomputed 80-vector candidate index, locally run `bge-reranker-base`, construct the prompt, and generate an answer. Ricci+BGE additionally applies the exact 20-step flow and retains a mean of 16.31 documents; BGE top-5 reranks all 20 and sends five. The total is therefore a deployment comparison rather than a matched-context mechanism test. Over 500 questions, Ricci+BGE takes 2.149 seconds on average and 4.436 seconds at p95, versus 1.365 and 3.242 seconds for BGE top-5, a 1.57× mean-latency ratio. The longer native Ricci context also increases generation latency, particularly on MultiHop-RAG.

Table 9: Flow-only runtime scaling for the sequential exact-transport audit implementation (500 queries per dataset, $M = 20$). Times exclude embedding API calls, reranking, and generation.

| Dataset | Mean seconds | | | Mean OT solves | | |
|---|---|---|---|---|---|---|
| | $k = 10$ | $k = 20$ | $k = 40$ | $k = 10$ | $k = 20$ | $k = 40$ |
| SQuADv2 | 0.080 | 0.301 | 1.416 | 646 | 2,379 | 8,954 |
| HotpotQA | 0.101 | 0.367 | 1.626 | 710 | 2,475 | 9,090 |
| MuSiQue | 0.086 | 0.319 | 1.482 | 702 | 2,493 | 9,182 |
| MultiHop-RAG | 0.070 | 0.283 | 1.370 | 588 | 2,268 | 8,757 |

Table 10: Full online latency and client-memory benchmark (500 questions per dataset). Times are seconds; memory columns are maximum client-process RSS and local PyTorch CUDA allocation in MB.

| Dataset | Path | Mean | p95 | Embed | Flow | BGE | GPT | CPU | GPU |
|---|---|---|---|---|---|---|---|---|---|
| SQuADv2 | Ricci+BGE dynamic | 1.946 | 2.982 | 0.203 | 0.771 | 0.204 | 0.740 | 1,657 | 1,387 |
| | BGE top-5 | 1.206 | 1.755 | 0.210 | 0 | 0.254 | 0.733 | 1,657 | 1,419 |
| HotpotQA | Ricci+BGE dynamic | 1.626 | 1.942 | 0.263 | 0.576 | 0.160 | 0.616 | 1,663 | 1,387 |
| | BGE top-5 | 1.046 | 1.782 | 0.224 | 0 | 0.193 | 0.622 | 1,663 | 1,419 |
| MuSiQue | Ricci+BGE dynamic | 1.513 | 1.950 | 0.193 | 0.564 | 0.121 | 0.625 | 1,664 | 1,339 |
| | BGE top-5 | 0.941 | 1.328 | 0.215 | 0 | 0.164 | 0.554 | 1,664 | 1,396 |
| MultiHop-RAG | Ricci+BGE dynamic | 3.513 | 6.162 | 0.216 | 0.290 | 0.272 | 2.696 | 1,738 | 1,369 |
| | BGE top-5 | 2.269 | 4.753 | 0.198 | 0 | 0.336 | 1.719 | 1,738 | 1,399 |
| All | Ricci+BGE dynamic | 2.149 | 4.436 | 0.219 | 0.550 | 0.189 | 1.169 | 1,738 | 1,387 |
| | BGE top-5 | 1.365 | 3.242 | 0.212 | 0 | 0.237 | 0.907 | 1,738 | 1,419 |

