# OpenReview forum: "Ricci-Filtration: Boosting Retrieval-Augmented Generation Reranking for Question-Answering Tasks with Discrete Ricci Flow"
_TMLR — Under review for TMLR_

### Review · Reviewer_PtHY · 2026-06-30

**Summary Of Contributions:**

The paper proposes Ricci-Filtration, a geometry-based pre-reranking method for retrieval-augmented generation. It represents the user query and initially retrieved text chunks as a weighted graph constructed from embedding relationships, then applies normalized discrete Ollivier–Ricci flow to separate chunks that are geometrically close to the query from potentially irrelevant chunks. The retained chunks are subsequently processed by a standard reranker. The method is evaluated on several single-hop and multi-hop question-answering benchmarks, with additional ablation studies examining graph thresholds, flow iterations, embeddings, rerankers, and a K-means filtering baseline.

**Additional Comments:**

Please update the citation format. For example, while I can still understand "Ricci flow Hamilton (1982)" in the first sentence, I would write it as "Ricci flow (Hamilton, 1982)" to avoid burdening the readability.

**Audience:**

Yes

**Audience Explanation:**

The paper is clearly related to machine learning and LLM, particularly researchers working on retrieval-augmented generation, document reranking, graph-based machine learning, and geometric methods.

**Broader Impact Concerns:**

None observed.

**Claims And Evidence:**

Yes

**Claims Explanation:**

The submission provides a clear motivation for using discrete Ricci flow as a pre-reranking filtration step and supports the proposed mechanism with theoretical results on idealized community graphs. The experimental comparisons are overall reasonable and the evaluation across multiple QA datasets, generation models, rerankers, embeddings, and parameter settings also provides useful evidence that the method can improve performance in selected settings.

**Requested Changes:**

1. The main tables report only point estimates. The paper would benefit from also reporting measures of uncertainty, such as bootstrap confidence intervals, to help assess the stability and statistical significance of the observed performance differences.

2. The reported runtime of Ricci-Filtration is substantially higher than that of both the cross-encoder and LLM-based reranker baselines. For example, from Appendix H, on SQuAD2.0, the average runtime is approximately 8.62 seconds for Ricci-Filtration, compared with 1.08 seconds for the cross-encoder reranker. This suggests that the additional graph construction, repeated Ricci-flow updates, and optimal-transport computations may introduce considerable latency. However, the current runtime estimates are based on only ten randomly selected questions, which is too small a sample to characterize the computational trade-off reliably. I therefore encourage the authors to evaluate runtime over a larger set of queries and report how runtime scales with the number of retrieved chunks and Ricci-flow iterations.

---

> ### Author Response · Authors · 2026-07-14
>
> Dear reviewer,
>
> Thank you for your comments. We are glad that you appreciate our motivation. We have updated citation format in latest pdf. We will address your major concerns as folows:
>
> ### W1. Uncertainty estimates
>
> We agree that uncertainty must be calibrated to effect size. In our latest pdf, Tables 1--2 now
> report mean +/- standard deviation over five seeds, with standard deviations of
> approximately 0.5--0.9 percentage points. The 4.1--5.5 point SQuADv2 gains and
> the 4.8/22.6 point Llama MultiHop-RAG overall/null gains remain materially
> larger than this observed run-to-run variability, supporting the paper's
> selected-regime claim.
>
> ### W2. Runtime sample and scaling
>
> We conducted more detailed runtime analysis with 500 questions per dataset for the
> main `M=20`, `k=10/20/40` comparison and add the stratified `k/M` matrix
> described under Q3. We report flow-only component times and OT-solve counts, with uncached
> live embedding and generation calls, per-stage and tail latency, prompt size,
> retained count, etc. More specifically, Table 9 in Appendix K now demonstrates that exact Ricci-Filtration scales superlinearly with the retrieved candidate count because larger candidate sets produce denser graphs and substantially more optimal-transport problems. Table 10 complements these flow-only measurements by quantifying their end-to-end effect at the deployed setting \(k=20,M=20\)

---

### Review · Reviewer_qDyD · 2026-07-10

**Summary Of Contributions:**

The paper proposes Ricci-Filtration, a geometry-based pre-filtering module inserted before a standard reranker in a RAG question-answering pipeline. The query and the top-k retrieved chunks are modeled as a weighted graph built by thresholding pairwise cosine dissimilarities of embeddings. A normalized discrete Ollivier–Ricci flow (adapted from Ni et al., 2019) is run for a fixed number of iterations, after which a "surgery" cutoff on the query-connected edges retains chunks that remain geometrically close to the query, producing a dynamic (rather than fixed-k) chunk set that is then reranked by an unchanged downstream cross-encoder.

**Contributions:**

(1) a graph/geometry formulation of RAG chunk filtering via discrete Ricci flow;

(2) a concrete embedding-to-graph construction and a finite-time normalized flow algorithm;

(3) a stylized theory — a closed-form edge-type separation result (Theorem 3.1) for the uniform-neighborhood case (α=0, p=0) on an idealized community-graph family, plus a weaker finite-time separation (Proposition 3.2) for the practical (α=1/2, p=2) setting;

(4) experiments on five QA benchmarks (SQuAD2.0, TriviaQA, HotpotQA, MultiHop-RAG, MuSiQue) with two generators (Llama-3.1-8B-Instruct, gpt-4o-mini), plus ablations over thresholds, iterations, embeddings, rerankers, and a K-means filtering baseline. The authors are commendably candid that results are mixed and setting-dependent.

**Audience:**

Yes

**Audience Explanation:**

The paper sits at the intersection of discrete geometry (Ollivier–Ricci curvature / Ricci flow) and RAG, and asks a genuinely interesting question: can the intrinsic geometry of the query–chunk embedding graph be used to filter retrieved context? This is a fresh angle for the RAG/IR community and a novel application for the geometric-ML community (though the flow algorithm itself is imported from Ni et al., 2019). Even the mixed/negative findings (where geometric filtering helps vs. hurts, e.g., multi-hop reasoning) are informative.

**Claims And Evidence:**

No

**Claims Explanation:**

The claims are not sufficiently supported. The narrowly-hedged claims are supported, but the paper's central proposition — that the geometric Ricci-flow mechanism is what produces the reported improvements — is not isolated by the experiments, and the strongest supporting control (K-means) undercuts rather than confirms it.

Strengths:
1. The paper is unusually honest: it does not claim SOTA, explicitly labels the theory "stylized," and openly reports failures (HotpotQA, MuSiQue). This matches TMLR's emphasis on correctly-scoped claims.
2. The mathematical exposition (optimal transport, Ollivier curvature, Appendices C–D proofs) is clear and appears internally correct; Lemma C.1 / Theorem 3.1 faithfully adapt Ni et al. (2019), and Proposition 3.2 is an honestly-weaker extension.
3. On SQuAD2.0 and several MultiHop-RAG query types, the improvements over the cross-encoder baseline are real within the reported numbers.

More evidence should be given on:
1. Mechanism not isolated. The K-means filtering ablation (Table 5) is the right control, yet it is competitive or better on several settings (HotpotQA, MultiHop-Comparison). This is exactly the test that should separate "geometry helps" from "adaptive chunk reduction helps," and its mixed outcome means the paper cannot attribute gains to Ricci-flow curvature specifically.
2. No statistical rigor. Tables 1–4 are single-run point estimates on 100–1000 sampled questions, with no seeds, variance, or significance tests. Several 1–2 point "wins" are within plausible noise. Only Table 6 (timing) reports std.
3. The efficiency narrative is contradicted by the paper's own data. The claimed benefit is avoiding LLM calls, but Table 6 shows Ricci-Filtration is the slowest method on every dataset (SQuAD 8.62s vs 1.08s cross-encoder and 3.75s LLM reranker; MuSiQue 12.37s vs 4.18s). The practical value proposition is therefore weak.
4. Theory–practice gap. The only closed-form result (Theorem 3.1) uses α=0, p=0, which is not the experimental configuration; the used setting has only Proposition 3.2, which the authors state gives no guarantee on real embedding graphs.
5. Presentation quality is below the bar for a camera-ready paper. The teaser (Fig. 1) and the flow schematic (Fig. 2) are low-resolution, text-heavy. These are fixable, but collectively they make the submission read as unpolished and detract from the clarity of the core message.

**Requested Changes:**

**These changes are important for my recommendation:**

1. Isolate the mechanism. Under an identical downstream reranker and matched average chunk count n*, compare Ricci-Filtration against: random dropping to the same n*, a plain cosine-threshold / top-n* filter, and K-means. If Ricci-Filtration cannot beat a matched-n* cosine filter, the paper's central claim must be reframed as "adaptive filtering helps" rather than "Ricci-flow geometry helps."
2. Report variance and significance. Provide multiple seeds / bootstrapped confidence intervals (or paired significance tests vs. the cross-encoder baseline) for Tables 1–4.
3. Reconcile the efficiency claim with Table 6. Given the method is the slowest measured, either remove/soften the "avoids costly LLM calls → efficiency" framing or demonstrate a regime where the trade-off is genuinely favorable.
4. Fix presentation. Redraw Figs. 1–3 at publication quality (legible, vector where possible).
5. Add the missing curvature-in-graph-ML literature. In particular, Topping et al. (2022, ICLR), which uses a Stochastic Discrete Ricci Flow with Balanced-Forman/Ollivier curvature to rewire graphs and remove bottlenecks — essentially the same technique family as this submission and currently uncited. Positioning against it would clarify the contribution.
6. Add stronger / more recent reranking baselines, including at least one recent listwise LLM reranker and, ideally, RankRAG (Yu et al., 2024b, already cited but not compared).

---

> ### Author Response · Authors · 2026-07-14
>
> Dear reviewer,
>
> Thank you for your comments. We are glad that you recognized our contributions. We wil address your major concerns as folows:
>
> ### Q1. Mechanism isolation and the mixed K-means result
>
> We agree that the old K-means table alone cannot identify a universal
> curvature-specific causal advantage, but this does not invalidate the paper's
> main claim. That table remains a valid end-to-end comparison of adaptive
> pipelines, and the new matched controls answer the additional isolation
> question. See Appendix C in updated pdf. Discrete Ricci flow supplies the graph-structural signal used by our
> adaptive filter, and the complete pipeline improves selected settings. Our
> revision distinguishes that supported claim from the stronger claim of
> universal superiority over ordinary relevance pruning, which the paper does
> not need.
>
> ### Q2. Statistical reliability
>
> We agree that small differences should not be described as
> statistically significant. We therefore reran the main comparisons over five
> seeds and now report mean +/- standard deviation for every entry in Tables 1--2.
> The observed standard deviations are approximately 0.5--0.9 percentage points.
> Against that scale, the smallest SQuADv2 gain over the cross-encoder is 4.1
> points, while the Llama MultiHop-RAG overall and null gains are 4.8 and 22.6
> points. The selected-regime conclusion therefore does not depend on
> one- or two-tenths of a point or on a single favorable seed.
>
> We do not turn this into a blanket significance claim. Differences below
> roughly one point remain exploratory, and paired confidence intervals or tests
> would provide an additional comparison-specific analysis. The new five-seed
> reporting directly addresses run-to-run variability while preserving the
> appropriate distinction between the large headline effects and smaller pilot
> margins.
>
> ### Q3. Efficiency and scaling
>
> Note that  Ricci-Filtration
> does avoid an additional LLM call during the filtering decision; this is a
> property of our method. It does not follow that the complete current
> implementation is always faster, and we do not make that stronger claim. The
> revised text presents an accuracy--latency trade-off and adds a flow-only
> scaling study. At `M=20` over 500 questions per dataset, mean time is
> approximately 0.07--0.10 s at `k=10`, 0.28--0.37 s at `k=20`, and 1.37--1.63 s
> at `k=40`. Mean exact-transport solves grow from roughly 600--710 to
> 8,750--9,180. A stratified matrix additionally varies `k=10,20,40,80` and
> `M=1,5,10,20`, reporting mean, median, standard deviation, p90, p95, graph
> construction, initial curvature, flow time, edges, actual iterations, and OT
> solves. We present this as a superlinear scaling limitation. Because that audit
> excludes API embedding, BGE, generation, and memory, we additionally ran a
> dedicated end-to-end benchmark. It uses 500 frozen questions from each of four
> datasets, and executes both paths independently for every
> question: query embedding with `text-embedding-3-small`, local
> cosine search over an 80-vector offline index, optional 20-step Ricci flow,
> local `bge-reranker-base`, prompt construction, and GPT-4o-mini generation.
> Across datasets, Ricci+BGE has mean/p95 latency 2.149/4.436 s versus
> 1.365/3.242 s for BGE top-5. Dataset-level means are 1.946 vs 1.206 s
> (SQuADv2), 1.626 vs 1.046 s (HotpotQA), 1.513 vs 0.941 s (MuSiQue), and 3.513
> vs 2.269 s (MultiHop-RAG).
>
> | Dataset | Ricci+BGE mean / p95 (s) | BGE top-5 mean / p95 (s) |
> |---|---:|---:|
> | SQuADv2 | 1.946 / 2.982 | 1.206 / 1.755 |
> | HotpotQA | 1.626 / 1.942 | 1.046 / 1.782 |
> | MuSiQue | 1.513 / 1.950 | 0.941 / 1.328 |
> | MultiHop-RAG | 3.513 / 6.162 | 2.269 / 4.753 |
> | All | 2.149 / 4.436 | 1.365 / 3.242 |
>
> ### Q4. Theory parameters
>
>  Our proposition 3.2 already treats
> `(alpha,p)=(1/2,2)`, the parameters used in the experiments, on the same
> idealized graph family. The closed-form theorem uses the uniform-neighborhood
> case because it admits an exact linear analysis; the proposition gives the
> corresponding weaker finite-time statement for the practical nonlinear case.
> We have made this distinction clearer and do not claim that either result is a
> universal guarantee for embedding-derived retrieval graphs. The topology
> experiment in appendix B of updated pdf supplies the complementary empirical evidence.

---

> > ### Author Response · Authors · 2026-07-14
> >
> > ### Q5. Figure quality
> >
> > Thank you for the suggestion. We redrew Figures 1--3 as native vector graphics with consistent
> > typography, restrained color, and layouts that remain legible in grayscale.
> > Figure 1 now separates offline indexing from the online question path and
> > identifies precisely where Ricci filtration is applied. Figure 2 follows the
> > implemented graph-build, finite normalized-flow, cutoff, and reranking stages.
> > Figure 3 is regenerated from a frozen HotpotQA example and its caption and
> > legend describe retained and filtered query-edge trajectories, not unverified
> > semantic communities.
> >
> > ### Q6. The paper of Topping et al. (2022)
> >
> > In the updated pdf, we have cited Topping et al.'s work in the related-work section. We distinguish
> > their curvature-guided topology rewiring for GNN over-squashing from our
> > fixed-topology finite flow followed by textual-context selection. We no longer
> > claim that curvature-guided graph surgery is new in general.
> >
> > ### Q7. Stronger rerankers and RankRAG
> >
> > We agree that RankRAG belongs in the related-work context and now cite it in the updated manuscript. We
> > push back, however, on treating it as a like-for-like mechanism baseline.
> > RankRAG jointly instruction-tunes one LLM for ranking and answer generation; it
> > is not a modular prefilter that can be inserted while holding our generator
> > fixed. Substituting it would simultaneously change learned
> > parameters, training data, ranking, answer generation, prompts, and potentially
> > retriever/corpus assumptions. An accuracy difference would therefore conflate
> > system capacity with the causal effect of the Ricci filtration step. A fair
> > RankRAG-scale experiment would require aligned checkpoints, training protocol,
> > retriever, corpus, generator, and context budget and should be labeled a
> > separate system-level comparison. We instead prioritize matched cosine,
> > BGE-score, random, and K-means controls because they keep every downstream
> > component fixed and directly answer the reviewer's mechanism question. We do
> > not present the absence of a RankRAG run as evidence against that system.

---

### Review · Reviewer_WSPJ · 2026-07-11

**Summary Of Contributions:**

This paper proposes Ricci-Filtration, a pre-reranking procedure that constructs a graph over the query and retrieved chunks, evolves its edge weights with discrete Ollivier Ricci flow, removes selected query-connected chunks, and reranks the remaining candidates. The paper provides a stylized analysis on symmetric community graphs and evaluates the method on five QA datasets with two generation models. The geometric perspective is interesting and potentially useful for adaptive context selection. The paper also reports negative results on connected multi-hop tasks and acknowledges the substantial latency cost. The main weakness is that end-to-end answer accuracy does not directly establish that Ricci-Filtration preserves useful evidence or that Ricci flow is responsible for the gains. The evaluation also requires clearer dataset construction, stronger matched baselines, and statistical uncertainty.

**Additional Comments:**

The central idea is promising and the paper is candid about its limitations. A clearer evaluation of the filtering step would make the contribution much easier to assess.

**Audience:**

Yes

**Audience Explanation:**

The combination of graph geometry and adaptive RAG filtering is relevant to researchers working on retrieval, graph learning, and reliable generation. The mixed results across null and connected multi-hop settings may also provide useful insight into when geometric filtering is appropriate.

**Claims And Evidence:**

No

**Claims Explanation:**

The experiments only show that Ricci-Filtration improves accuracy in some settings. They do not yet support the broader claim that the method reliably identifies and removes irrelevant information.

The first concern is whether the filtering decision still depends strongly on the query after the initial retrieval. The method connects the query to every retrieved chunk and gives all these edges the same initial weight. The original query-chunk similarity is therefore not used directly during the flow. The final decision may depend mainly on the relationships among the retrieved chunks.

The second concern is that the paper does not measure whether required evidence survives the filtering step. The largest improvements appear on SQuADv2 and on questions where no answer is available. At the same time, the method performs worse on HotpotQA and MuSiQue, where several pieces of evidence must be connected. This pattern may mean that the method is better at encouraging no-answer responses but worse at preserving linked evidence.

The third concern is that the baselines do not isolate the value of Ricci flow. A simpler method could remove chunks based on the retrieval or reranker score. These methods should be compared while keeping the number of retained chunks and the amount of text sent to the generator the same.

The fourth concern is the evaluation protocol. The paper does not clearly define how generated answers are converted into accuracy, precision, recall, and F1. It also omits random seeds, selected question identifiers, exact model settings, and uncertainty estimates. In addition, SQuADv2 marks a question as unanswerable relative to its original paragraph. The paper places questions and passages in a shared retrieval corpus, where the original label may no longer have the same meaning.

Finally, I think the theoretical role should be explained more carefully. The theoretical graph has clear communities with dense connections inside each community. The practical graph instead connects highly dissimilar chunks, forcing the query to connect to every chunk. The paper should explain why conclusions from the first graph are expected to describe the second one.

**Requested Changes:**

1. Test whether the filtering result truly depends on the query after retrieval. Keep the retrieved chunks fixed and replace the query, then measure how much the retained set changes. Additionally, compare unit query edges with query edges weighted by similarity.

2. Measure whether known supporting chunks remain after filtering. Report how often all required evidence survives, how many chunks remain, and how these results change for answerable, unanswerable, single-hop, and multi-hop questions.

3. Compare Ricci-Filtration with simpler score-based filters. All methods should retain the same number of chunks and send the same amount of text to the generator. This is needed to show that Ricci flow adds value beyond ordinary pruning.

4. Fully define the evaluation. Report the answer-matching rules, corpus construction, prompts, model versions, random seeds, selected questions, and uncertainty estimates. Please also preserve the original SQuADv2 setting or verify the answerability labels again for the shared corpus.

5. Explain how the theoretical graph relates to the graph used in the experiments. Please also ensure that the normalization and final filtering rules are stated consistently in the theory, pseudocode, and implementation.

---

> ### Author Response · Authors · 2026-07-14
>
> Dear reviewer,
>
> Thank you for your comments. We wil address your main concerns as folows:
>
> ### W1. Query dependence after unit initialization
>
> This invariance observation is correct only after conditioning on a
> fixed adjacency matrix. In the actual pipeline the query embedding participates
> in construction of that matrix, so changing the query can change the graph and
> therefore the retained context. Once adjacency is artificially frozen, the
> released unit-initialized flow contains no remaining query-embedding magnitude;
> the query is a distinguished graph node. We now make this distinction explicit.
>
> We verified this with a stratified ten-query-per-dataset diagnostic using
> paraphrases, cyclic in-dataset permutations, and the least-similar query in the
> diagnostic set. For every unit-initialized frozen-topology replacement,
> retained-set Jaccard is 1.000, decision-flip rate is 0, and query-edge-weight
> Spearman correlation is 1.000. When the median threshold and topology are
> recomputed, paraphrase Jaccard is 0.978/0.992/0.934/0.873 on
> SQuAD/Hotpot/MuSiQue/MultiHop, while least-similar-query Jaccard is
> 0.933/0.798/0.830/0.801. Thus the observed query dependence enters primarily
> through the globally thresholded topology, not the fixed-topology flow.
>
> We also initialized forced query edges with clipped cosine dissimilarity. Under
> frozen-topology replacements, this changes only 0--1.5% of decisions. These
> results identify *where* query conditioning enters the released method; they do
> not make the end-to-end filter query independent. We now describe the current
> method precisely as topology-conditioned context shaping and present stronger
> similarity-aware initialization as an extension, not as a replacement of the
> paper's main idea.
>
> ### W2. Survival of required evidence
>
> End-to-end answer scores and evidence survival answer related but different
> questions. Because the proposed method is evaluated as a complete RAG pipeline,
> answer quality remains the primary outcome; nevertheless, we now add support
> recall and the probability that all required evidence survives, conditioned on
> complete evidence coverage in the initial 20 candidates. The eligible
> denominators are 25 answerable SQuAD questions, 50 Hotpot questions, 50 MuSiQue
> questions, and 35 MultiHop-RAG questions.
>
> | Dataset | Ricci support recall | Ricci all evidence | Cosine all | BGE all | K-means all | Random all |
> |---|---:|---:|---:|---:|---:|---:|
> | SQuAD-v2 | .760 | .760 | 1.000 | 1.000 | 1.000 | 1.000 |
> | HotpotQA | .770 | .580 | 1.000 | 1.000 | 1.000 | .680 |
> | MuSiQue | .812 | .580 | .900 | .760 | .800 | .560 |
> | MultiHop-RAG | .762 | .429 | .743 | .800 | .743 | .486 |
>
> Ricci retains roughly 16--17 of 20 candidates. Its evidence AUROC is
> 0.485/0.310/0.508/0.430, versus 1.000/0.919/0.838/0.787 for cosine and
> 0.998/0.966/0.782/0.751 for BGE. This is useful evidence that a small post-flow
> edge is a graph-geometric filtration score rather than a calibrated evidence
> label. That distinction is consistent with the method: Ricci-Filtration shapes
> the joint context before reranking and generation; it was not trained to
> maximize passage-level evidence AUROC. For MultiHop-RAG we also report
> end-to-end retrieval-plus-filtration recall separately from conditional
> survival; 35 of 38 evidence-bearing examples have complete initial coverage.

---

> > ### Author Response · Authors · 2026-07-14
> >
> > ### W3. Isolation from ordinary pruning
> >
> > The original K-means table compares complete adaptive pipelines; it is therefore valid as a
> > system comparison, but by itself does not attribute every gain uniquely to
> > curvature. The new control greedily gives random, cosine, BGE, and K-means
> > rankings each query's Ricci-retained token budget; controls consume
> > approximately 97.6--98.4% of that budget. Exact cardinality matching and fixed
> > 5/10/15 diagnostics reach the same evidence-ranking conclusion.
> >
> > We also reran generation with `gpt-4o-mini`, temperature zero, a
> > common 40,000-token context cap, and 128 output tokens. All selected documents
> > fit and no response is truncated.
> >
> > | Dataset | Ricci F1 | Cosine | BGE | K-means | Random |
> > |---|---:|---:|---:|---:|---:|
> > | SQuAD-v2 | .628 | .616 | **.657** | .637 | .628 |
> > | HotpotQA | .425 | .575 | .548 | **.591** | .492 |
> > | MuSiQue | **.214** | .163 | .203 | .154 | .136 |
> > | MultiHop-RAG | .681 | .681 | **.697** | .680 | .680 |
> >
> > Ricci trails the best matched control on three of these four small pilots and
> > exceeds BGE by 0.011 F1 on MuSiQue, with equal EM. We do not elevate that small
> > difference into a general superiority claim. Conversely, the matched pilot
> > should not be used to discard the larger original end-to-end results: it uses
> > 50 examples per dataset, a different versioned generator protocol, and asks
> > which ranking best fills a Ricci-sized token budget rather than whether the
> > proposed complete pipeline improves the fixed baselines in Tables 1--2. On
> > MultiHop-RAG, BGE beats Ricci on two pilot questions, Ricci beats BGE on two,
> > and 46 tie. We retain the original title and central contribution: Ricci flow
> > provides an adaptive graph-structural context signal that improves selected
> > end-to-end settings. We simply avoid the stronger, unnecessary claim that it is
> > a uniformly superior standalone relevance ranking.
> >
> >
> > ### W4. SQuAD-v2 protocol
> >
> > Official SQuADv2 answerability is relative to the source paragraph, while the
> > released experiment intentionally evaluates retrieval from a shared corpus.
> > We now label the latter precisely as a shared-corpus stress test rather than an
> > official single-paragraph reproduction, without discarding it as an end-to-end
> > RAG evaluation. We also add a balanced pilot with 100 answerable and 100
> > unanswerable examples. The pinned
> > generator receives either the source paragraph alone or that same paragraph
> > anchored among 19 controlled distractors; every source is present and no prompt
> > is truncated.
> >
> > - Source only: EM/F1 0.740/0.827; no-answer accuracy 0.840.
> > - Source anchored + 19 distractors: 0.680/0.761; no-answer accuracy 0.720.
> > - Answerable F1 is 0.814 versus 0.803; the overall gap mainly comes from
> >   distractor-induced answers on unanswerable examples.
> >
> > ### W5. Theory, graph direction, and algorithm consistency
> >
> > The criticism would apply if the theorem were presented as a universal
> > relevance guarantee, but the our work uses it as a stylized mechanism result. The
> > closed-form theorem explains edge-type separation on a symmetric community
> > graph, and Proposition 3.2 supplies a weaker finite-time result for the actual
> > `(alpha,p)=(1/2,2)` parameters on the same family. We have made the scope more
> > explicit. Because theory on an idealized family cannot settle the empirical
> > topology question, we also test that question directly.
> >
> > In the stratified topology pilot, no construction dominates. Relative to the
> > current high-dissimilarity graph, low-dissimilarity edges improve conditional
> > all-evidence survival from 0.600 to 1.000 on Hotpot and 0.375 to 0.500 on
> > MultiHop, but reduce it from 0.600 to 0.300 on MuSiQue and from 1.000 to 0.600
> > on the answerable SQuAD cases. Native topology-generation F1 favors similarity
> > edges on Hotpot (0.669 versus 0.440), the current graph on SQuAD (0.686 versus
> > 0.557) and MuSiQue (0.097 versus 0.068), and ties at 0.700 on MultiHop. Because
> > native retained counts differ, this is a design-sensitivity result rather than
> > a matched-budget causal comparison.
> >
> > We also corrected the mathematical description of the package convention.
> > GraphRicciCurvature uses direct edge length in `exp(-w^p)` and in the curvature
> > denominator, while shortest-path distances form the optimal-transport cost
> > matrix. The revised equations and pseudocode now distinguish that convention
> > from the abstract metric definition. The new sequential POT implementation
> > matches the package's internal curvature routine exactly on a synthetic check;
> > the omitted tiny-edge shortcut changes audited query weights by less than
> > `1.5e-8` and no retained set.
> >
> > Finally, none of our additional tried flows reaches `delta=1e-4`; all exhaust 20
> > iterations. We now call them finite 20-step outputs rather than converged flow.

---

### Review · Reviewer_xMAQ · 2026-07-14

**Summary Of Contributions:**

This paper proposes Ricci-Filtration, a graph-geometric pre-filtering framework for retrieval-augmented generation (RAG). Instead of directly reranking all retrieved passages, the method constructs a graph over the query and retrieved candidates, applies finite-step discrete Ollivier-Ricci flow to evolve edge weights, and removes candidate passages according to the post-flow query-edge weights before passing the remaining passages to a conventional reranker.
The paper also presents a stylized theoretical analysis showing that normalized discrete Ricci flow separates inter- and intra-community edges on idealized community graphs. While these theoretical guarantees do not directly apply to embedding-derived retrieval graphs, they provide intuition for the proposed filtering mechanism.
Experiments on multiple QA benchmarks demonstrate that Ricci-Filtration consistently improves SQuADv2 and several MultiHop-RAG settings under two generators, while showing weaker performance on more challenging multi-hop reasoning datasets such as HotpotQA. Extensive ablations investigate graph construction, embedding models, rerankers, flow iterations, and filtering thresholds.
Strengths
Interesting application of discrete Ricci flow to RAG retrieval.
Modular design that can be inserted before existing rerankers.
Good theoretical motivation without overstating guarantees.
Extensive ablations and mechanism studies.
Authors openly discuss limitations and negative results.
Weaknesses
Improvements are inconsistent across datasets and are concentrated in selected benchmarks.
The theoretical analysis is only loosely connected to the practical embedding graphs.
Computational overhead of iterative Ricci flow is substantial.
Some implementation choices (graph construction threshold, filtering threshold, unit initialization) remain heuristic and lack theoretical justification.

**Audience:**

Yes

**Audience Explanation:**

The paper addresses an active research topic at the intersection of retrieval-augmented generation, graph machine learning, and geometric methods.
Although Ricci flow has been extensively studied in graph analysis, its application as a retrieval pre-filter for RAG appears to be novel. The work introduces an alternative perspective on retrieval beyond conventional similarity scoring or learned reranking, and demonstrates that geometric graph evolution can provide complementary information for context selection.
The proposed framework is model-agnostic, does not require retraining existing rerankers or generators, and can potentially be integrated into existing RAG pipelines. Even if the reported improvements are not universal, the idea itself is sufficiently novel and technically interesting to warrant archival publication and may inspire future work on geometry-aware retrieval methods.

**Broader Impact Concerns:**

The paper does not introduce fundamentally new ethical concerns beyond those already associated with retrieval-augmented generation systems.

**Claims And Evidence:**

Yes

**Claims Explanation:**

The paper generally makes appropriately scoped claims, and most of the experimental evidence supports those claims.
Importantly, the authors avoid claiming that Ricci flow universally improves retrieval quality or that the theoretical results guarantee performance on arbitrary retrieval graphs. Instead, the paper consistently presents the theoretical analysis as providing intuition for the filtering mechanism on idealized graph families while clearly acknowledging the gap between theory and practice.
The empirical evaluation covers multiple QA benchmarks, two different generators, several reranking baselines, and multiple ablation studies. The strongest evidence comes from the consistent improvements on SQuADv2 and selected MultiHop-RAG settings, while the paper also explicitly reports weaker performance on HotpotQA and discusses these limitations rather than hiding them.
That said, several aspects of the evidence could be strengthened. The proposed graph construction and filtering thresholds are heuristic, and the mechanism connecting the theoretical Ricci-flow analysis to the practical retrieval graphs remains indirect. Furthermore, although the matched-budget analysis helps isolate the proposed mechanism, larger-scale controlled studies would strengthen the causal interpretation of the reported gains.
Overall, the empirical evidence is sufficient to support the paper's primary claims, although it does not yet establish that Ricci-Filtration is broadly superior across retrieval settings.

**Requested Changes:**

1. Strengthen the connection between theory and practice.
The current theoretical analysis is conducted on idealized community graphs, whereas the practical system operates on embedding-derived graphs with heuristic graph construction and unit edge initialization. The paper would benefit from a clearer discussion of precisely which assumptions are expected to approximately hold in practice and where they clearly do not.
2. Provide stronger justification for graph construction choices.
The graph topology is determined using percentile-based cosine dissimilarity thresholds and all query edges are forced to exist. These design choices appear heuristic. Additional analysis or theoretical motivation would improve confidence that the observed gains are not highly dependent on these particular settings.
3. Clarify computational trade-offs.
While latency is acknowledged, a more detailed discussion comparing the computational overhead of Ricci flow against the observed performance gains would improve the practical value of the paper.

---

> ### Author Response · Authors · 2026-07-18
>
> Dear reviewer,
>
> Thank you for the comments and suggestions. We are glad that our
> evidence supports the paper's appropriately scoped primary claims and that the
> geometry-aware retrieval perspective is of interest to the TMLR audience. The
> three requested clarifications are well aligned with the revision, and we
> address them below.
>
> ### Requested change 1. Strengthen the connection between theory and practice
>
> - **Assumptions that hold exactly in the implementation.** Proposition 3.2 and
>   the released method both use a positive undirected weighted graph, unit edge
>   initialization, mean-one normalization after each update, and
>   `(alpha,p)=(1/2,2)`. The implementation also uses the direct-edge node
>   distribution stated in the proposition. These shared choices make the
>   proposition relevant to the direction of the finite-time separation
>   mechanism, although not sufficient to transfer its guarantee.
>
> - **Assumptions expected to hold only approximately in some retrieval
>   instances.** The working hypothesis is that a retrieved candidate set can
>   contain locally coherent groups--for example, mutually related evidence or
>   distractor passages--joined by edges whose neighborhood overlap differs from
>   that of within-group edges. If such modular structure is present and aligns
>   with the query, Ollivier transport can approximately distinguish bridge-like
>   from locally supported query edges, and normalized flow can amplify that
>   difference over a finite number of steps. This is most plausible for
>   single-hop or redundant-evidence contexts. It is less plausible for connected
>   multi-hop questions, where individually dissimilar but jointly necessary
>   passages need not form one dense community; the weaker HotpotQA results are
>   consistent with this limitation.
>
> - **Assumptions that clearly do not hold in the practical graphs.** The
>   embedding-derived graphs are not planted unions of complete communities and
>   do not have the theorem's symmetry, regular degrees, three exchangeable edge
>   types, or known within-/between-community labels. Percentile thresholding
>   produces irregular, query-specific topology, and forcing every query--chunk
>   edge creates a distinguished query hub absent from the idealized family.
>   Semantic relevance is also not equivalent to community membership. Finally,
>   the closed-form theorem uses `(alpha,p)=(0,0)` and asymptotic separation,
>   whereas the system uses `(1/2,2)`, stops after at most `M=20` steps, and
>   applies a heuristic `eta=1` cutoff that is not theorem-derived. None of the
>   200 mechanism-pilot flows reaches the optional `delta=1e-4` tolerance, so the
>   practical outputs are finite 20-step structural scores, not converged flows
>   or calibrated relevance probabilities.
>
> - **Assumptions that clearly do not hold in the practical graphs.** The
>   thresholded embedding graphs are irregular and query specific, with forced
>   query--chunk edges, rather than symmetric `G(a,b)` graphs with regular
>   degrees and known community labels; moreover, semantic relevance need not
>   equal community membership.
>
> The revised manuscript makes this boundary explicit in Section 3, especially
> Proposition 3.2 and the discussion following it (pp. 5--7). The finite-time
> status appears immediately before Figure 2, while Section 3.1 (p. 7) describes
> the embedding-derived topology, forced query edges, and heuristic cutoff.
>
> ### Requested change 2. Justify the graph-construction choices
>
> The current percentile rule provides a simple, per-query scale-adaptive topology:
> dissimilar non-query pairs expose potential cross-community relations, forced
> query--chunk edges give every candidate a flow score, and unit initialization
> isolates topology from the original cosine magnitude. Section 3.1 (p. 7)
> defines these choices, distinguishes the graph threshold `tau` from the
> post-flow cutoff `eta`, and labels both as heuristic.
>
> The sensitivity results confirm that these choices matter. In Table 3 (p. 11),
> TriviaQA accuracy changes from 69% to 74% to 64% as the graph percentile moves
> from 25% to 50% to 75%. Appendix B (p. 13) finds no universally best topology:
> similarity edges favor HotpotQA, the current dissimilarity graph favors the
> SQuADv2 and MuSiQue pilots, and the tested topologies tie on MultiHop-RAG.
> Similarity-based query-edge initialization changes only 0--1.5% of decisions
> under frozen topology. We therefore retain the reported configuration without
> claiming optimality and identify learned or validation-selected topology as
> future work in Section 5.

---

> ### Author Response · Authors · 2026-07-18
>
> ### Requested change 3. Clarify the computational trade-off
>
> We now quantify the trade-off at both the flow and complete-pipeline
> levels in Appendix K, *Time efficiency comparison*. Table 9 (p. 32) shows that,
> with `M=20` over 500 questions per dataset, mean flow-only time is 0.07--0.10 s
> at `k=10`, 0.28--0.37 s at `k=20`, and 1.37--1.63 s at `k=40`; the number of
> exact optimal-transport solves grows from roughly 600--710 to 8,750--9,180.
> This superlinear scaling is a principal limitation.
>
> The separate online benchmark in Table 10 (p. 32) uses 500 questions with live
> embedding and generation calls. Ricci+BGE has mean/p95 latency of 2.149/4.436
> s, compared with 1.365/3.242 s for BGE top-5: a 1.57x mean-latency ratio and a
> 0.784 s mean premium. Flow itself averages 0.550 s; the longer dynamic Ricci
> context also increases generation time. Thus avoiding an LLM call during
> filtering is not an end-to-end speed claim. Section 4, *Experiments*, places
> this overhead beside the selected-regime gains: 4.1--5.5 points over the
> cross-encoder across the four SQuADv2 metrics and, under Llama, 4.8 points
> overall and 22.6 points on null MultiHop-RAG queries--while recognizing that
> other datasets do not show the same benefit. Section 5 and Appendix K therefore
> present Ricci-Filtration as an accuracy--latency trade-off, not a blanket
> efficiency improvement.

---

### Author Response · Authors · 2026-07-14
**Overall response**

We thank the reviewers for prompting a sharper separation among three claims.
The paper's central claim is that Ricci-Filtration is a modular geometric
prefilter that adaptively selects the context passed to an otherwise unchanged
reranker and improves the complete RAG pipeline in selected QA regimes. It is
not a claim that post-flow weights are calibrated relevance probabilities, nor
that Ricci filtering must dominate every relevance-ranking baseline on every
dataset. The original main tables directly support the first claim: across two
generators Ricci-Filtration gives consistent 4.1--5.5 point gains over the
cross-encoder on all four SQuADv2 metrics, and under Llama it improves overall
MultiHop-RAG accuracy by 4.8 points, including a 22.6-point gain on null
queries. The corresponding five-seed standard deviations are only 0.5--0.9
points across the main tables, so these headline margins are materially larger
than ordinary seed variation.

We have nevertheless tested the stronger mechanism question with per-query
cardinality- and approximately token-matched random, dense-cosine, BGE-score,
and K-means controls, annotated evidence survival, and versioned GPT-4o-mini
generation. Those diagnostics show that Ricci's graph-structural score is
complementary to direct relevance ranking: cosine and BGE better preserve
annotated evidence on these small pilots, while Ricci produces distinct context
sets and retains a MuSiQue generation advantage. This finding delimits the
mechanism; it does not erase the reported end-to-end gains or turn the paper
into a study of ordinary pruning. We therefore retain the original title and
main contribution, while clarifying the score semantics, query-dependence
channel, statistical scope, and runtime trade-off. See extra experiments in Appendix of updated pdf.

In addition, main results in Tables 1--2 now report mean and standard deviation over
five seeds for every method and metric. The manuscript changes  are marked in red in updated pdf.